# Pretrained Vision-Language-Action Models are Surprisingly Resistant to Forgetting in Continual Learning

**Huihan Liu** [1]   **Changyeon Kim** [1 2]   **Bo Liu** [3]   **Minghuan Liu** [1]   **Yuke Zhu** [1]

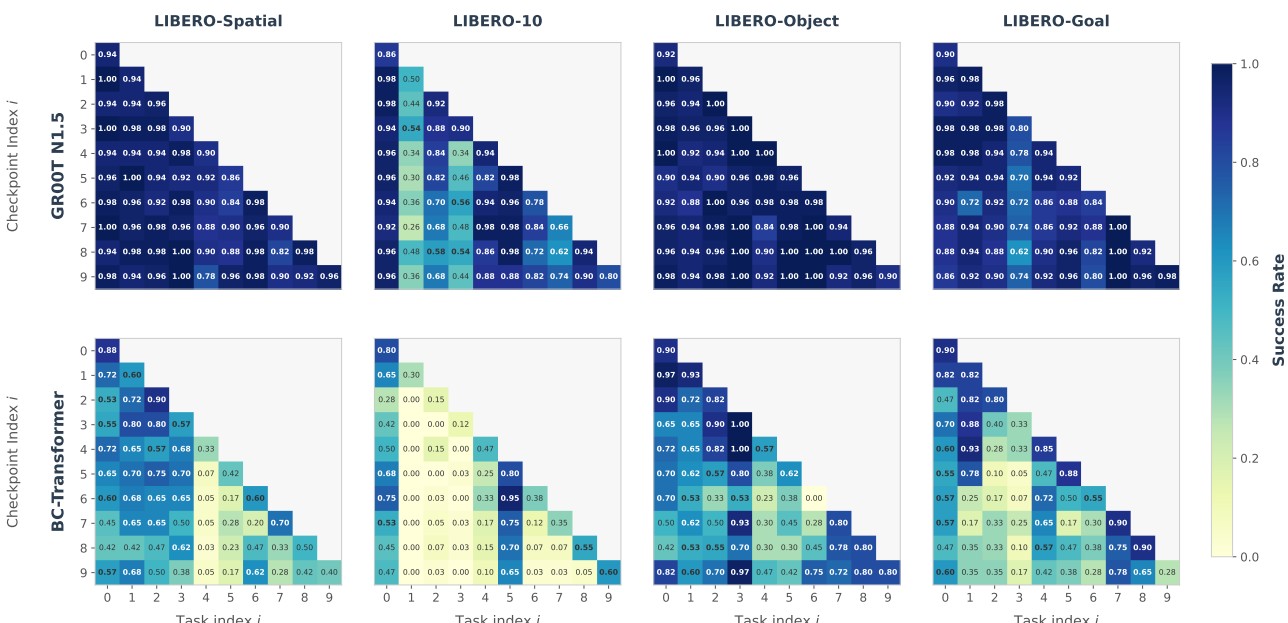

*Figure 1.* **Comparison of continual learning performance between a pretrained Vision-Language-Action (VLA) model (`GR00T N1.5`; NVIDIA et al. (2025)) and a non-pretrained small policy model (`BC-Transformer`; Liu et al. (2023b)).** Each checkpoint corresponds to a model obtained by sequentially training over ten tasks under Experience Replay (ER), where the parameters at the start of training for checkpoint $i$ are initialized from checkpoint $i-1$. Each matrix entry $(i, j)$ denotes the success rate on Task $j$ after training on Task $i$. The columns track how a given task performance evolves as training continues (top to bottom). We compare a pretrained VLA model (top) with a non-pretrained small BC policy (bottom) across multiple LIBERO benchmark suites.

## Abstract

Continual learning is a long-standing challenge in robot policy learning, where a policy must acquire new skills over time without catastrophically forgetting previously learned ones. While prior work has extensively studied continual learning in relatively small behavior cloning (BC) policy models trained from scratch, its behavior in modern large-scale pretrained Vision-Language-Action (VLA) models remains underexplored. In this work, we found that pretrained VLAs are re-
markably resistant to forgetting compared with smaller policy models trained from scratch. Simple Experience Replay (ER) works surprisingly well on VLAs, sometimes achieving zero forgetting even with a small replay data size. Our analysis reveals that pretraining plays a critical role in downstream continual learning performance: large pretrained models mitigate forgetting with a small replay buffer size while maintaining strong forward learning capabilities. Furthermore, we found that VLAs can retain relevant knowledge from prior tasks despite performance degradation during learning new tasks. This knowledge retention enables rapid recovery of seemingly forgotten skills through finetuning. Together, these insights imply that large-scale pretraining fundamentally changes the dynamics of continual learning, enabling models to continually acquire new skills

[1]The University of Texas at Austin [2]KAIST [3]Microsoft Superintelligence. Correspondence to: Huihan Liu <huihanl@utexas.edu>.

*Proceedings of the $43^{rd}$ International Conference on Machine Learning*, Seoul, South Korea. PMLR 306, 2026. Copyright 2026 by the author(s).

over time with simple replay. Code and more information can be found at project website.

# 1. Introduction

Continual learning (French, 1999a; McCloskey & Cohen, 1989a; Liu et al., 2023b) remains a long-standing challenge in robot policy learning, which requires robot policies to acquire new skills over time while preserving previously learned behaviors. In contrast to learning multi-task policies at once (Levine et al., 2016; Jang et al., 2022a; Brohan et al., 2023b;a), continual learning requires the policy to balance the ability of learning new tasks (plasticity) and preserving previously acquired knowledge (stability) (French, 1999a; McCloskey & Cohen, 1989a). The learning is non-trivial, as the policy tends to fail and result in catastrophic forgetting (French, 1999a; McCloskey & Cohen, 1989a; Kirkpatrick et al., 2017a; Luo et al., 2025), where the learning of new skills often leads to large degradation in previously learned behaviors, making naive sequential finetuning ineffective (Shenfeld et al., 2025).

Prior work on continual learning in robotics has primarily focused on relatively small policy models trained from scratch or with limited pretraining (Liu et al., 2023b; Chaudhry et al., 2019; Kirkpatrick et al., 2017a; Mallya & Lazebnik, 2018; Wan et al., 2024). In these settings, catastrophic forgetting is often pervasive (Liu et al., 2023b), and people typically require the use of large replay buffers (Chaudhry et al., 2019) or carefully designed regularization techniques (Kirkpatrick et al., 2017a) for mitigation. Recently, large-scale pretrained Vision-Language-Action models (VLAs) have demonstrated strong generalization and downstream task transfer in robotic manipulation. Pretrained on diverse multimodal data (Radford et al., 2021; Touvron et al., 2023; OpenAI et al., 2024) and robot trajectories, these models rely less on task-specific parameter updates and instead adapt by reusing and reconfiguring existing representations. This raises a central question: do large pretrained VLAs behave differently in continual learning from the smaller models, and if so, in what ways?

Our answer follows by observing that pretrained VLAs exhibit vastly different dynamics from smaller policy models in continual learning. In particular, we conduct a comprehensive empirical study that reveals a set of unexpected dynamics of VLAs in continual learning:

**Pretrained VLAs are remarkably resistant to forgetting compared with smaller policy models trained from scratch (Sec. 3).** Specifically, we found that simple Experience Replay (ER) works surprisingly well on VLAs, often achieving *zero* forgetting even with a small replay data size (*e.g.*, 2% of training data in LIBERO benchmark suites). Furthermore, sometimes it even enables ***positive backward***

***transfer*** on previously learned tasks, *i.e.*, further improving the performance of previous tasks with only the replay data stored. These dynamics of learning have not been seen before on smaller models.

**Pretraining plays an integral role in improving continual learning performance in both forward and backward transfer (Sec. 5).** We found that the pretrained knowledge in VLAs is especially effective for mitigating forgetting even at a small replay dataset size (*e.g.*, 2% of training data). Furthermore, pretraining reduces forgetting while still maintaining a high success rate on new tasks, avoiding the trade-off between preserving knowledge and forward transfer ability.

**VLAs preserve relevant knowledge of prior tasks despite performance degradation from learning new ones (Sec. 6).** Even when performance on previous tasks seems to degrade from catastrophic forgetting, the underlying knowledge is still preserved in the VLA's internal representation. Evidence for this lies in the fact that a few finetuning steps can quickly restore performance on past tasks.

# 2. Related Work

## 2.1. Continual Learning Beyond Training from Scratch

Catastrophic forgetting has long been a central challenge in continual learning, motivating a wide range of algorithmic solutions such as regularization, distillation, replay, and architectural isolation (McCloskey & Cohen, 1989b; French, 1999b; Kirkpatrick et al., 2017b; Li & Hoiem, 2016; Lopez-Paz & Ranzato, 2017; Rusu et al., 2016). Most prior work studies these mechanisms in relatively small models trained from scratch, revealing a fundamental stability-plasticity trade-off and a strong dependence on replay size and task order (Parisi et al., 2019; De Lange et al., 2022).

More recently, continual learning has been revisited in the context of large pretrained models, particularly language models. Empirical studies show that pretrained representations can reduce—but not eliminate—forgetting under sequential finetuning, and that pretraining substantially alters transfer and interference patterns (Wu et al., 2022; Scialom et al., 2022). Complementary work on continual pretraining and temporal benchmarks further highlights the tension between knowledge retention and adaptation in large models (Jang et al., 2022b;c; Hu et al., 2023). These results suggest that conclusions drawn from training-from-scratch settings may not directly generalize to large pretrained models.

## 2.2. VLAs and Lifelong Robot Learning

Vision-language-action models (VLAs) extend foundation model paradigms to robotics by jointly learning perception, language understanding, and control from large, het-

erogeneous data (Reed et al., 2022; Brohan et al., 2022; Zitkovich et al., 2023; Driess et al., 2023). Recent open efforts such as Open X-Embodiment, Octo, and OpenVLA demonstrate strong transfer across tasks and embodiments, making VLAs natural candidates for lifelong robot learning (Open X-Embodiment Collaboration et al., 2023; Octo Model Team et al., 2024; Kim et al., 2025).

Continual learning in robotics has traditionally focused on smaller behavior cloning policies and task-specific settings (Lesort et al., 2020; Liu et al., 2023a). While iterative adaptation frameworks suggest that generalist policies can improve over time (Bousmalis et al., 2023; Yadav et al., 2026), the mechanisms by which large pretrained VLAs retain or recover previously learned skills under continual finetuning remain poorly understood. In particular, it is unclear whether classical replay-based conclusions—such as the need for large buffers or explicit regularization—still apply in the regime of large pretrained VLAs.

**Positioning of this work.** In contrast to prior studies, we systematically investigate continual learning behavior in modern pretrained VLAs. We show that simple experience replay surprisingly achieves near-zero forgetting with surprisingly small buffers, and that large-scale pretraining fundamentally changes the stability–plasticity dynamics observed in smaller models trained from scratch. Our findings position pretrained VLAs as a distinct continual learning regime, where forgetting, transfer, and recovery are governed more by pretraining and representation reuse than by specialized continual learning algorithms.

## 3. Preliminaries

### 3.1. Continual Learning in Robotics

We model a robotic task as a finite-horizon Markov Decision Process: $\mathcal{M} = (\mathcal{S}, \mathcal{A}, \mathcal{T}, H, \mu_0)$, where $\mathcal{S}$ and $\mathcal{A}$ are the state and action space, $\mu_0$ is the initial state distribution, and $\mathcal{T} : \mathcal{S} \times \mathcal{A} \to \mathcal{S}$ is the transition function. To check the success of the given task, we assume access to a goal predicate $g : \mathcal{S} \to \{0, 1\}$. Under this formulation, continual learning, also known as lifelong learning (Liu et al., 2023b), considers a robot that sequentially learns $K$ tasks $\{T^k\}_{k=1}^{K}$ using a single task-conditioned policy $\pi(\cdot \mid s, T)$. Each task $T^k \equiv (\mu_0^k, g^k)$ is specified by a unique initial-state distribution $\mu_0^k$ and a goal predicate $g^k$, while sharing the same $\mathcal{S}, \mathcal{A}, \mathcal{T}, H$ across tasks.

In this work, we focus on continual learning via imitation learning. Formally, we assume access to expert demonstrations for each task $T^k$, denoted by $D^k = \{\tau_i^k\}_{i=1}^{N}$, where each trajectory $\tau_i^k = \{(o_0, a_0), \ldots, (o_{l_k}, a_{l_k})\}$ consists of observations $o_t$ (robot sensory inputs) and actions $a_t$, terminating at $l_k \leq H$. After observing tasks up to $T^k$, our goal

is to obtain a policy $\pi$ satisfying the following objective:

$$\min_{\pi} \ J_{\text{BC}}(\pi) = \frac{1}{k} \sum_{p=1}^{k} \mathbb{E}_{(o_t, a_t) \sim D^p} \left[ \sum_{t=0}^{l_p} \mathcal{L}\big(\pi(o_{\leq t}; T^p), a_t\big) \right],$$

where $\mathcal{L}$ denotes the behavior cloning loss. We assume that demonstrations from previous tasks $\{D^p : p < k\}$ are not fully available when learning $T^k$.

**Experience Replay (ER).** Experience Replay (Chaudhry et al., 2019) is a widely used approach in continual learning that retains a subset of samples from previous tasks and leverages them to learn new tasks. After completing policy learning for a task, ER stores a portion of that task's data in a separate replay buffer. When training on a new task, ER draws samples from the buffer and combines them with the current task's training data, so that the effective training distribution better approximates the empirical distribution over all tasks seen so far.

### 3.2. Vision-Language-Action Models

Vision-Language-Action models (VLAs) are robotic policies that map visual observations and natural language instructions to actions, typically by building on top of large vision-language models (VLMs) pretrained on internet-scale image-text data. To adapt VLMs for robotic control, VLAs are commonly further trained via imitation learning on large-scale robot demonstration datasets with diverse embodiments $\mathcal{D}$ (Liu et al., 2025). Formally, a VLA defines a policy $\pi$ that, at each time step $t$, predicts an action chunk $\mathbf{a}_t$ conditioned on a language instruction $l$ and a history of observations $\mathbf{o}_{\leq t}$ of length $H$. Each observation typically includes multi-view RGB images $\mathbf{I}_t^1, \ldots, \mathbf{I}_t^n$ and the proprioceptive state $\mathbf{q}_t$. The VLA encodes these inputs with its VLM backbone to obtain a latent representation, which is then used by an action head to predict future actions.

## 4. VLAs are Surprisingly Resistant to Forgetting

### 4.1. Evaluating VLAs in Continual Learning

To find out if VLAs behave differently in continual learning from the smaller models, we evaluated VLAs in a set of continual learning experiments.

**Benchmarks and Datasets.** For our experiments, we employ LIBERO (Liu et al., 2023b), a lifelong learning benchmark for robotic manipulation. We use four different task suites (LIBERO-Spatial, LIBERO-10, LIBERO-Object, LIBERO-Goal) to validate the knowledge transfer across different properties. We use the same task ordering for all experiments. For the training dataset, we use the filtered dataset introduced by Kim et al. (2024).

*Table 1.* **Continual learning performance on LIBERO benchmarks**: Average Success Rate (SR ↑) and Negative Backward Transfer (NBT ↓). Results are with sample size = 1000 (20% of dataset size per task), in accordance with LIBERO's official setting. `BC-DP`, `BC-T`, `BC-ViT` are abbreviations for `BC-Diffusion Policy`, `BC-Transformer`, `BC-ViT` respectively.

| Method | LIBERO-Spatial | | LIBERO-Object | | LIBERO-Goal | | LIBERO-10 | | Average | |
|---|---|---|---|---|---|---|---|---|---|---|
| | SR (↑) | NBT (↓) | SR (↑) | NBT (↓) | SR (↑) | NBT (↓) | SR (↑) | NBT (↓) | SR (↑) | NBT (↓) |
| `Pi0` | $0.879 \pm 0.008$ | $0.019 \pm 0.019$ | $0.897 \pm 0.011$ | $-0.011 \pm 0.034$ | $0.732 \pm 0.015$ | $-0.005 \pm 0.010$ | $0.563 \pm 0.028$ | $-0.068 \pm 0.018$ | $0.768 \pm 0.017$ | $-0.016 \pm 0.022$ |
| `GR00T` | $0.940 \pm 0.013$ | $0.007 \pm 0.009$ | $0.975 \pm 0.004$ | $0.019 \pm 0.013$ | $0.943 \pm 0.004$ | $0.023 \pm 0.017$ | $0.820 \pm 0.017$ | $0.059 \pm 0.035$ | $0.919 \pm 0.011$ | $0.027 \pm 0.021$ |
| `BC-DP` | $0.663 \pm 0.074$ | $0.050 \pm 0.070$ | $0.847 \pm 0.069$ | $0.025 \pm 0.056$ | $0.809 \pm 0.089$ | $0.230 \pm 0.101$ | $0.464 \pm 0.024$ | $0.201 \pm 0.044$ | $0.696 \pm 0.068$ | $0.127 \pm 0.071$ |
| `BC-T` | $0.659 \pm 0.058$ | $0.299 \pm 0.096$ | $0.595 \pm 0.112$ | $0.132 \pm 0.120$ | $0.709 \pm 0.022$ | $0.356 \pm 0.042$ | $0.376 \pm 0.034$ | $0.192 \pm 0.019$ | $0.585 \pm 0.066$ | $0.245 \pm 0.080$ |
| `BC-ViT` | $0.513 \pm 0.069$ | $0.171 \pm 0.065$ | $0.543 \pm 0.268$ | $0.077 \pm 0.134$ | $0.661 \pm 0.024$ | $0.319 \pm 0.022$ | $0.316 \pm 0.062$ | $0.204 \pm 0.063$ | $0.508 \pm 0.142$ | $0.193 \pm 0.082$ |

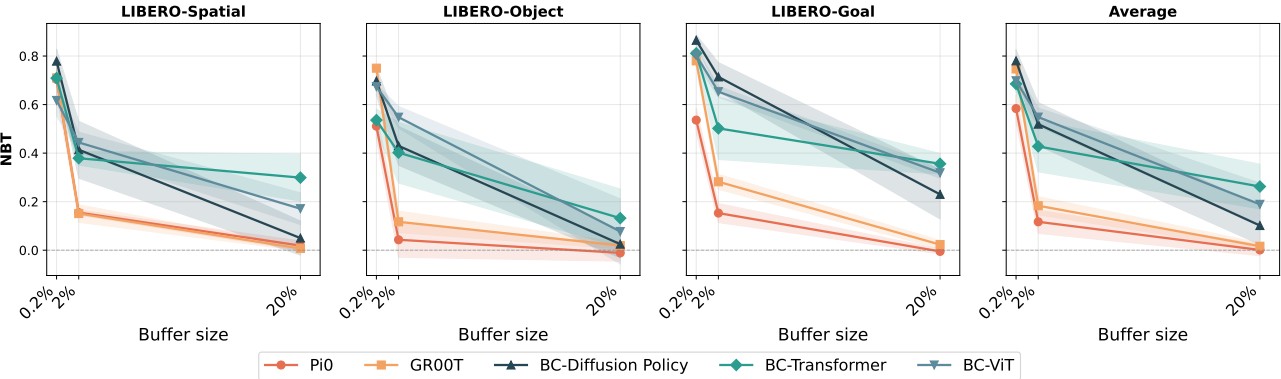

*Figure 2.* **Negative Backward Transfer (NBT) across different replay buffer sizes.** Each subplot shows NBT as a function of replay buffer size ($\{0.2\%, 2\%, 20\%\}$) for all methods across the four benchmarks and their average. Shaded regions indicate $\pm 1$ standard deviation across seeds. Higher NBT indicates more forgetting; values near zero indicate no forgetting. Results and discussion for `LIBERO-10` are reported separately in Tab. 6 in Appendix A.2.

**Models.** To verify that our hypothesis holds robustly across diverse VLA backbones, we consider `Pi0` (Black et al., 2026) and `GR00T N1.5` (NVIDIA et al., 2025) which differ in architecture, parameter count, and pretraining data. For the non-pretrained small policy, we adopt `BC-Transformer` (Liu et al., 2023a).

**Training.** We train a separate model for each task suite. Within each suite, we train the model for a fixed number of optimization steps per stage while carrying over the model weights across tasks. For ER, at task $k$, we train on the current task dataset, together with a fixed replay buffer containing $M$ randomly sampled transitions per task from previous tasks (reusing the same stored samples throughout). Unless otherwise specified, we use $M = 1000$.

**Evaluation.** We report two metrics: average task success rate (SR) (%), and negative backward transfer (NBT) based on success rate, following Liu et al. (2023a). Let $c_{i,k}$ denote the success rate (%) on task $i$ when evaluated using the agent trained up to task $k$. For each $k \in [1, 2, \cdots, K]$, we evaluate $\{c_{i,k}\}_{i=1}^k$, yielding the collection $\{\{c_{i,k}\}_{i=1}^k\}_{k=1}^K$. We compute NBT as follows:

$$\text{NBT} = \frac{1}{K}\sum_{k=1}^K \text{NBT}_k, \text{NBT}_k = \frac{1}{K-k}\sum_{\tau=k+1}^K \left(c_{k,k} - c_{k,\tau}\right).$$

Intuitively, NBT measures how much past knowledge is

*lost* after learning new tasks. A positive NBT indicates knowledge loss; the lower the value, the better.

### 4.2. The Surprising Effectiveness of Experience Replay

As shown in Tab. 1, we observed that pretrained VLAs trained with experience replay (ER) surprisingly achieve ***near-zero, or even positive*** backward transfer across multiple LIBERO benchmarks, while simultaneously maintaining a strong forward transfer. The positive backward transfer observed in models like `Pi0` and `GR00T N1.5` indicates that for pretrained VLAs, learning new tasks can sometimes even improve the performance on previously learned ones, which challenges the prior understanding of the stability-plasticity trade-off (McCloskey & Cohen, 1989a; French, 1999a). This behavior contrasts with smaller policy models, which typically suffer severe forgetting under comparable continual learning settings. Extended details of this experiment can be referred to in Appendix B.1.

**There are consistent trends across different VLAs.** We observed similarly strong resistance to forgetting across multiple VLAs (`Pi0`, `GR00T N1.5`) despite differences in architecture, parameter count, and pretraining recipes. This consistency suggests that the effectiveness of ER is a general property of pretrained VLAs, rather than an artifact of any particular architecture design or pretraining dataset mixture.

*Table 2.* Comparison with non-ER continual learning baselines (Sequential, EWC) on LIBERO benchmarks.

| Model | Method | LIBERO-Object | | LIBERO-10 | |
|---|---|---|---|---|---|
| | | SR (↑) | NBT (↓) | SR (↑) | NBT (↓) |
| Pi0 | Sequential | 0.910 | 0.696 | 0.644 | 0.562 |
| | EWC | 0.910 | 0.608 | 0.622 | 0.543 |
| | ER | 0.898 | **-0.007** | 0.586 | **-0.070** |
| GR00T | Sequential | 0.964 | 0.752 | 0.852 | 0.758 |
| | EWC | 0.826 | 0.766 | 0.816 | 0.728 |
| | ER | 0.962 | **0.004** | 0.836 | **0.082** |

*Table 3.* **Continual learning on real robot**. Each lower-triangular entry $(i, j)$ is the success rate (%) on task $j$ after sequential training through task $i$ under experience replay with Pi0.

| After training on task | Evaluate on task | | |
|---|---|---|---|
| | $T_0$ | $T_1$ | $T_2$ |
| $T_0$ | 44.4 | — | — |
| $T_1$ | 47.1 | 29.4 | — |
| $T_2$ | 44.4 | 28.6 | 33.3 |

**VLAs are resistant to forgetting, especially under a low replay data regime.** Fig. 2 shows a clear difference in how pretrained VLAs and non-pretrained baselines utilize replay data across different buffer sizes. When the buffer size is 2% (100 samples per task), pretrained VLAs such as Pi0 and GR00T still have low NBT around 0.1–0.2, while non-pretrained baselines (BC-Transformer, BC-ViT, and BC-Diffusion Policy) deteriorate to 0.4–0.5, retaining 2–4× more forgetting. The non-pretrained models require substantially more replay data ($> 20\%$) to achieve comparable NBT. Note that under the smallest buffer size (0.2%), the gap appears less pronounced in absolute NBT; however, this is largely a limitation of the metric itself: at such a small buffer, all policies experience near-complete forgetting, and the absolute NBT is dominated by each policy's initial success rate rather than its true resistance to forgetting. A normalized variant of NBT that accounts for this effect is discussed in Appendix A.2.

**ER is uniquely effective.** To further validate the effectiveness, we compare ER against baselines that do not explicitly re-use past-task data and instead train using only the current task dataset. We consider two such variants: (i) Sequential (Liu et al., 2023b), which simply carries over the model weights across tasks and continues finetuning on the current task data without any replay buffer, and (ii) EWC (Kirkpatrick et al., 2017a), which likewise trains on the current task data but adds a regularization penalty to discourage changes to parameters important for earlier tasks. As shown in Tab. 2, neither baseline consistently improves over non-VLA policies in reducing forgetting, whereas ER remains robust for pretrained VLAs. These results indicate that explicitly revisiting past-task data during training (even if it's a little) is crucial for preserving (and sometimes improving) prior task performance.

**Real-Robot Experiments.** Our main experiments study continual learning in simulation (LIBERO). To test whether the observation generalizes to real robots, we run a small-scale experiment with the same protocol on the real-robot task suite from Mutex (Shah et al., 2023), which provides tabletop manipulation tasks with distinct object layouts and language goals in the home-kitchen workspace (Franka arm

with multi-view RGB and proprioceptive inputs). We sequentially train Pi0 on three tasks with different visual layouts: (T0) Pick up bread and place it on the plate; (T1) Pick up the pink cup and place it on the plate; (T2) Pick up the red cup and place it in the back caddy. After each stage, we evaluate all tasks seen so far using the same success criteria as in simulation and report results in Tab. 3. The average SR = 35.4% and NBT = −0.003, indicating negligible backward transfer. The qualitative trend matches our simulation findings: pretrained Pi0 with simple ER can learn new skills without substantially erasing previously learned ones.

Together, these results establish that VLAs are resistant to forgetting with ER, enabling strong forward transfer with little to no forgetting across architectures. Naturally, we wonder: what are the underlying dynamics behind this?

## 5. Pretraining Plays an Integral Role in Improving Continual Learning Performance

In this section, we investigate a defining characteristic of VLAs—large-scale pretraining—and analyze how it interacts with experience replay across different replay sizes. Our analysis reveals that large-scale pretraining is a key factor affecting both forward and backward transfer behavior. Specifically, we find *pretraining i) reduces forgetting especially in low-data regimes and ii) preserves knowledge of past tasks while still maintaining strong forward transfer*, which will be elaborated below.

**Methodology.** To isolate the role of pretraining, we conduct controlled comparisons of VLAs finetuned under the same architecture, differing only in the amount of pretraining during model initialization. In particular, we compare three variants: i) Pi0 from VL + Action, the Pi0 (Black et al., 2026) model pretrained on large-scale robot datasets on top of a VLM (PaliGemma; Beyer et al. (2024)) backbone; ii) Pi0 from VL, with Pi0 initialized from only PaliGemma VLM backbone without robot data pretraining; and iii) Pi0 from scratch, where the model uses Pi0 architecture but is trained entirely from scratch. For each variant, we run continual learning with experience replay and vary the replay buffer size to examine how continual learning behaviors

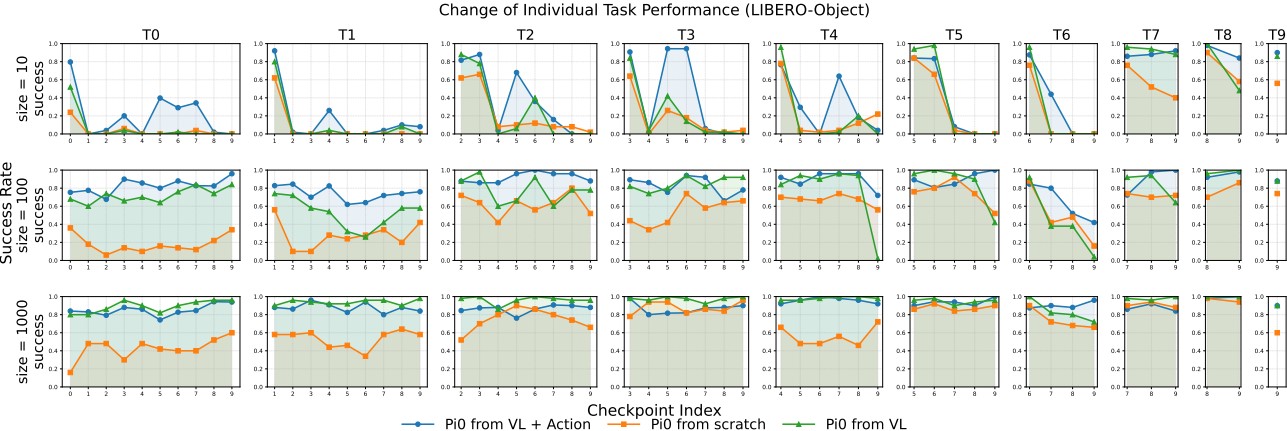

*Figure 3.* Comparison of forgetting performance across different buffer sizes (10, 100, 1000) for `Pi0` pretrained, `Pi0` initialized from Paligemma, and `Pi0` trained from scratch.

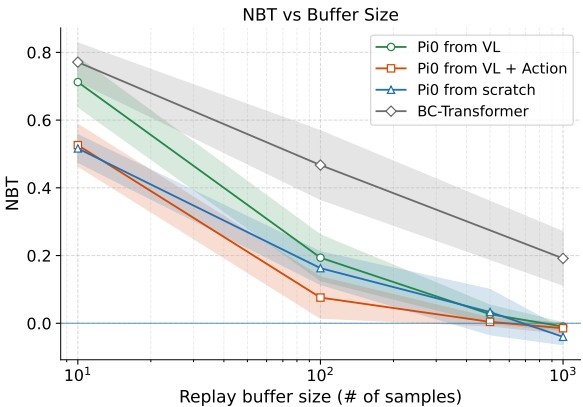

*Figure 4.* Pareto frontier of average NBT vs. replay buffer size. We compare the forgetting performance (lower is better) across different buffer sizes for `Pi0` model with different levels of pretraining. We also provide `BC-Transformer` as a non-pretrained, smaller model reference.

change with different replay buffer sizes.

**Pretrained knowledge is useful for mitigating forgetting, especially *under small replay buffer size*.** Fig. 4 visualizes this effect through a Pareto frontier that characterizes the trade-off between forgetting (in terms of Negative Backward Transfer) and replay buffer size. Specifically, when the curve reaches 0 in the y-value, it means there is zero forgetting; when it reaches below 0, it means there is some positive transfer to past tasks (past tasks improve performance after training on later tasks). As the replay buffer size decreases, the gap between pretrained and non-pretrained models increases, indicating that at a smaller replay buffer size, pretraining plays an increasingly important role in mitigating forgetting.

While the Pareto frontier provides a summary of forgetting as a function of replay size, scalar forgetting metrics alone

can obscure important dynamics during continual learning. In particular, they do not capture how performance on individual tasks degrades or recovers as new tasks are learned. To make these dynamics explicit, we visualize how the success rate of each task $k$ ($T_k$) changes after learning Task $k$, $k + 1, \ldots$, in Fig. 3. We show this for replay buffer size $= 10, 100, 1000$ respectively for `LIBERO-Object`. We found several interesting dynamics:

- When the replay buffer size is relatively large (1000 samples, about 20% of the full dataset): in this data regime, there is little to no forgetting for all variants. However, the pretrained variants give a much higher success rate in learning new tasks and maintaining performance.

- When the replay buffer is small with only 10 transitions (about 0.2% of full dataset): while all variants have forgetting, `Pi0` from VL + Action displays less forgetting than the other two variants, and could still recover partial performance of some tasks (*e.g.*, T2, T3, T4).

The above observation implies an interesting fact: investigating forgetting itself can be limited, because a policy where task success rates are consistently low might give the same backward transfer with one where task success rates are consistently high. This brings us to the second observation:

**Pretraining mitigates forgetting while still maintaining high forward transfer.** Tab. 4 shows that both `Pi0` from VL + Action and `Pi0` from VL achieve consistently higher success rate than the model trained from scratch (sample size = 1000). The result suggests that pretraining improves continual learning not only by reducing forgetting, but also by enabling forward learning, avoiding the degenerate outcome where low forgetting arises from insufficient plasticity, as also noted in concurrent work (Hu et al., 2026).

*Table 4.* Comparison of continual learning metrics for different levels of pretraining on LIBERO benchmarks.

| Method | LIBERO-Spatial | | LIBERO-Object | | LIBERO-Goal | | Average | |
| | SR (↑) | NBT (↓) | SR (↑) | NBT (↓) | SR (↑) | NBT (↓) | SR (↑) | NBT (↓) |
|---|---|---|---|---|---|---|---|---|
| `Pi0` from VL + Action | 0.889 | -0.00013 | 0.9179 | 0.0009 | 0.782 | -0.0974 | 0.863 | -0.0322 |
| `Pi0` from VL | 0.9199 | 0.00469 | 0.946 | 0.0105 | 0.832 | 0.0325 | 0.899 | 0.0159 |
| `Pi0` from scratch | 0.528 | -0.0795 | 0.694 | -0.0301 | 0.7419 | -0.0082 | 0.655 | -0.0393 |
| `BC-Transformer` | 0.590 | 0.180 | 0.725 | 0.101 | 0.720 | 0.293 | 0.678 | 0.191 |

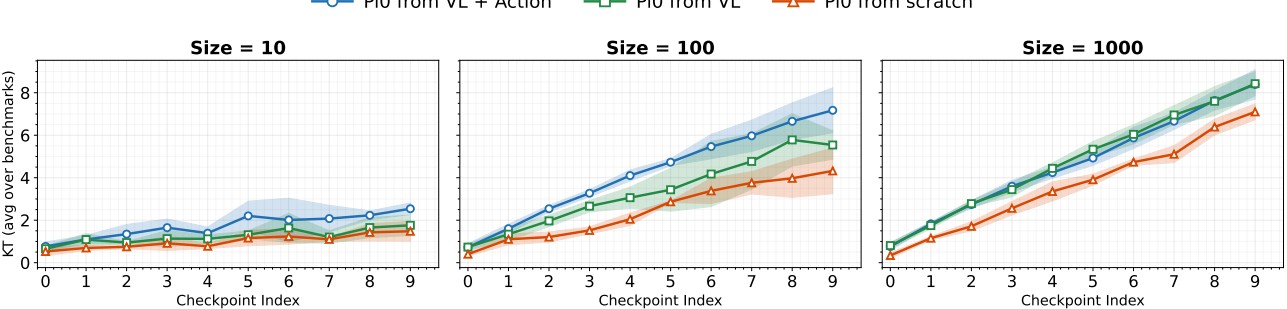

*Figure 5.* **Knowledge transfer** (sum of success rates) curves across four benchmarks. We compare `Pi0` trained from scratch (orange), `Pi0` trained from PaliGemma (green), and `Pi0` pretrained (blue) under different replay buffer sizes (10, 100, 1000).

While backward transfer quantifies how well previously learned tasks are preserved, it does not capture whether a model continues to effectively acquire new tasks over time. In particular, low measured forgetting can arise either from genuine knowledge retention or from insufficient plasticity, where the model fails to meaningfully learn new tasks. To give a more balanced view of stability and plasticity, we additionally analyze *knowledge transfer* (KT), which measures the aggregate success rate across all tasks and thus reflects the overall aggregated learning progress throughout continual training (see Fig. 5). Higher slope reflects more accumulation of the task knowledge.

We examine how knowledge transfer evolves throughout continual learning under different replay buffer sizes. Fig. 5 compares this aggregate knowledge transfer for the three variants using the `Pi0` model. We observed that `Pi0` from VL + Action and `Pi0` from VL exhibit steadily increasing knowledge transfer over time, indicating effective learning of new tasks while preserving prior knowledge. In contrast, `Pi0` from scratch often displays slower growth in knowledge transfer. This shows that low measured forgetting in non-pretrained models can arise from an inability to learn new tasks, rather than genuine knowledge preservation.

Finally, beyond pretraining, we conduct experiments to assess whether other factors may also influence continual learning behavior in VLAs, in particular model size and training objective. We show the results in Appendix C.

## 6. VLAs Retain Knowledge that is Seemingly Forgotten

In this section, we present our key insight that *a drastic decrease in prior task performance does not necessarily indicate complete forgetting of task-relevant knowledge in pretrained VLAs*; instead, this knowledge is still retained, as prior task performance can often be rapidly recovered with minimal finetuning.

**Methodology.** We begin by investigating which types of knowledge are most susceptible to performance degradation when pretrained VLAs are trained under a continual learning setting. To this end, we decompose the model's knowledge into three components—vision, language, and action—corresponding to representations stored in the vision-language (VL) backbone and the action head. We adopt the following methodology, illustrated in Fig. 6:

- **Baseline** (Fig. 6 (a)): The model learns three consecutive tasks (Task $k-1$, $k$, and $k+1$). As the model moves from Task $k$ to Task $k+1$, we measure the extent to which learning Task $k+1$ degrades performance on Task $k$.

- **Component swapping** (Fig. 6 (b)): To identify which part of the model is responsible for the forgetting—the VL backbone or the action head—we swap components between different stages of training. Specifically, we combine the VL backbone learned for Task $k+1$ with the action head from Task $k$ and evaluate performance on Task $k$. A drop in the model's performance on Task $k$

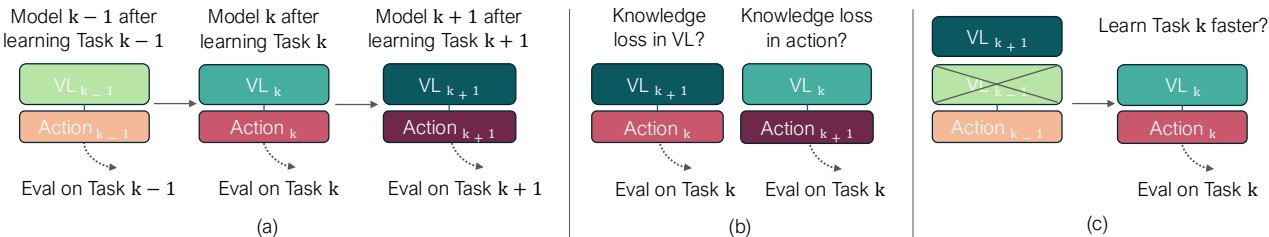

(a)            (b)            (c)

*Figure 6.* **Methodology for investigating VLA knowledge loss in continual learning.** *(a)*: Continual learning pipeline of learning Task $k - 1$, Task $k$ and Task $k + 1$ sequentially. *(b)*: Identifying knowledge loss in the vision-language (VL) backbone and action head by swapping components of Task $k$ and Task $k + 1$. *(c)*: Recovering vision-language knowledge with finetuning by learning on Task $k$ with backbone from Task $k + 1$. If the model preserves knowledge from Task $k$ after learning Task $k + 1$, using Task $k + 1$'s VL backbone should make learning Task $k$ faster compared with learning Task $k$ for the first time from the VL backbone of $k - 1$.

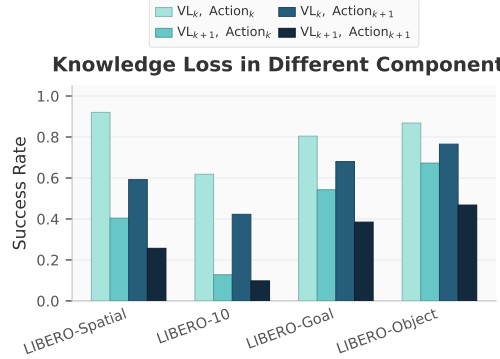

*Figure 7.* **Comparison of mean success rates retained under different vision-language and action combinations.**

indicates that the backbone has lost relevant VL knowledge. Conversely, we pair the VL backbone from Task $k$ with the action head trained after Task $k + 1$; degraded performance in this case is attributed to forgetting in the action head.

- **Task performance recovery** (Fig. 6 (c)): This experiment investigates whether the task knowledge in a given component (*e.g.*, the VL backbone) is still retained after learning a new task. Using the VL backbone obtained after Task $k + 1$, we retrain the model on Task $k$ and measure the learning speed. If the model reaches peak performance faster than during its original training on Task $k$ (which started from Task $k - 1$ backbone), this suggests that the knowledge is partially retained. Otherwise, comparable or slower relearning indicates the knowledge has likely been completely overwritten.

**What knowledge do VLAs appear to forget?** We conducted component swapping with `Pi0` backbone under replay buffer size $= 10$ (0.2%), where forgetting is high as shown in the last section. We made three interesting observations:

*Table 5.* **Recovery efficiency.** *Finetuning* steps (as ratio of the original training time) needed to regain the peak success rate achieved when learning the task *for the first time*.

| Benchmark | Recovery steps in ratio ↓ | |
| --- | --- | --- |
| | `Pi0` | `BC-Transformer` |
| `LIBERO-Spatial` | **0.066** | 1.36 |
| `LIBERO-10` | **0.105** | 1.87 |
| `LIBERO-Object` | **0.067** | 1.80 |
| `LIBERO-Goal` | **0.062** | 0.33 |

- Knowledge is compartmentalized across VLA components: Swapping either the action head or the VL backbone results in performance that is consistently worse than the original $(\text{VL}_k, \text{Action}_k)$ model, yet better than the fully updated $(\text{VL}_{k+1}, \text{Action}_{k+1})$ model. This pattern suggests that knowledge loss in VLA is not monolithic, but instead affects different modules separately.

- VL backbone is the dominant source of forgetting: Across all four task categories, swapping the VL backbone $(VL_{k+1})$ leads to a larger performance drop compared to swapping only the action head $(A_{k+1})$. This indicates that action-relevant information is relatively more consistent across tasks, whereas updates to the VL backbone alter the underlying representations, leading to a mismatch with previously learned action mappings and consequently greater performance degradation.

- Knowledge loss correlates with task diversity: The degree of performance degradation aligns with the extent of task variation. For example, `LIBERO-10`, which features the most diverse visual backgrounds, exhibits the largest drop when swapping $\text{VL}_{k+1}$. In contrast, `LIBERO-Object` involves similar pick-and-place action across different objects, so it shows minimal degradation when swapping $\text{Action}_{k+1}$.

**VLAs preserve relevant knowledge of prior tasks despite performance degradation from learning new ones.**

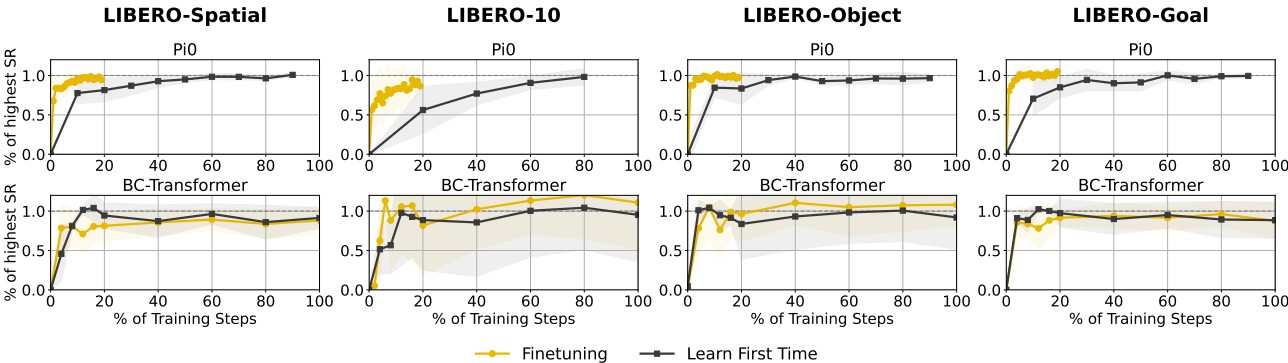

*Figure 8.* **Performance comparison averaged across tasks.** Each column represents a different Libero benchmark (Spatial, 10, Object, Goal) where the x-axis shows the percentage of training steps (0–100%) during finetuning. `Pi0` results stop at 20% of the training steps, while `BC-Transformer` stops at 60% of the training steps.

Next, we zoom in to examine whether the knowledge in the VL backbone is still preserved using finetuning as a probing tool. For finetuning, we use the same dataset (entire dataset for the task) with its first training, as well as the same model architecture and optimization scheme. For this experiment, we use `Pi0` backbone and we compare it with `BC-Transformer`.

Fig. 8 shows that `Pi0` reaches peak performance within a small fraction of training steps, significantly faster than its initial training. This indicates that task knowledge is preserved in the model and reused during relearning. In contrast, `BC-Transformer` requires a similar number of steps as initial training to reach peak performance, suggesting that task knowledge has been largely erased.

Tab. 5 quantifies this behavior using *recovery efficiency*, defined as the ratio between the steps required to recover the peak success rate ($T_f$) and the original training steps ($T_o$). Across all benchmarks, `Pi0` consistently recovers peak performance with fewer than $10\%$ of the original training steps, whereas `BC-Transformer` often requires a comparable or greater number of steps and sometimes exhibits unstable finetuning. These results indicate that, although pretrained VLAs may exhibit apparent forgetting in task-level performance (as shown in NBT), the underlying task knowledge is often retained in their internal representations and can be rapidly re-expressed through limited finetuning. On the other hand, non-pretrained models tend to relearn tasks largely from scratch because prior knowledge is erased.

## 7. Conclusion and Discussion

In this paper, we evaluate VLAs in continual learning and demonstrate that they are surprisingly resistant to forgetting. To understand the underlying dynamics, we conduct comprehensive experiments on large-scale pretraining and on how task-relevant knowledge is retained. Our findings

shed light on future design of continual learning paradigms for VLAs: 1) Continual learning for large VLAs may not require increasingly complex algorithms; instead, it may benefit from strong pretraining and small replay data. 2) To mitigate forgetting, future works should consider effectively reusing knowledge retained in VLA representations, rather than relying solely on larger replay buffers.

**Limitations.** Most of our evaluation is in simulation on LIBERO (Liu et al., 2023b), where all suites share the same Franka embodiment and similar tabletop environments; we have not tested continual learning extensively across different robots, simulators, or large cross-benchmark shifts. While we report a small real-robot pilot study ($K{=}3$ tasks, single `Pi0` run), broader hardware studies with multiple seeds, baselines, and diverse conditions are needed before transferring continual learning setups to real-world deployment. Furthermore, our continual learning protocol depends on experience replay over stored teleoperation transitions. Replay buffers can become costly as tasks accumulate, and may be infeasible under privacy, consent, or data-retention constraints in real-world deployments. Lastly, while the forgetting comparisons span multiple VLAs and BC baselines, our deeper mechanistic studies—including pretraining ablations, component swapping, and recovery analyses—focus mainly on `Pi0`. Future work should extend these analyses to other VLA architectures to determine which mechanisms of forgetting and recovery are model-specific or broadly shared.

## Acknowledgment

We thank the Texas Advanced Computing Center (TACC) and the Center for Generative AI at UT Austin for providing valuable computing resources. We thank Rutav Shah and Yu Lei for their valuable discussions and helpful feedback on the manuscript, and Abhiram Maddukuri for sharing technical details and codebase setup tips. We thank Zihui Xue and Xixi Hu for providing helpful guidance on the TACC compute setup. We thank Yuqi Xie for providing helpful guidance on the `GR00T N1.5` codebase. We thank Zhiyuan Zhou and Yajat Yadav for sharing the customized pretrained `Pi0` model checkpoints. This work was partially supported by the National Science Foundation (FRR-2145283, EFRI-2318065), the Office of Naval Research (N00014-24-1-2550), the DARPA TIAMAT program (HR0011-24-9-0428), and the Army Research Lab (W911NF-25-1- 0065). It was also supported by the Institute of Information & Communications Technology Planning & Evaluation (IITP) grant funded by the Korean Government (MSIT) (No. RS-2024-00457882, National AI Research Lab Project).

## Impact Statement

**Applications.** This work sits at the intersection of continual learning methods and Vision-Language-Action (VLA) models for robotic manipulation. By characterizing forgetting, experience replay, and knowledge retention in large pretrained VLAs, our results may influence how researchers and practitioners design lifelong imitation-learning pipelines. For example, by relying more on strong pretraining and modest replay rather than on increasingly complex continual-learning algorithms developed for smaller behavior-cloning models. Potential applications include more data-efficient sequential skill acquisition with smaller replay buffers and faster recovery of prior skills via rapid fine-tuning when policy performance drops.

**Implications.** These application trends could have both positive and negative societal consequences, though the magnitude and timing are uncertain. On the positive side, more stable continual learning may lower the cost of maintaining strong manipulation policies across long task sequences, which could eventually support assistive or industrial robotics where systems must accumulate skills over time. On the negative side, improved continual learning for generalist manipulation policies may accelerate automation and labor-market disruption; we do not study distributional or economic effects here. Our use of demonstration replay also raises privacy and consent concerns when trajectories come from human operators. Large-scale VLA pretraining carries substantial compute and environmental costs, which our scaling analysis touches on but does not fully quantify at deployment scale. A further risk is over-trust in low measured forgetting or rapid recovery: policies can appear stable on held-out evaluation while still failing on long-tail rare conditions, which is especially consequential in real-world manipulation settings.

**Initiatives.** We encourage follow-up work that (i) develops continual-learning metrics beyond task success, for example, including diagnostics for latent retention failures and sensitivity to task ordering; (ii) studies fairness, consent, and safe handling of replayed demonstrations; and (iii) tracks the environmental footprint of scaling VLA pretraining alongside downstream continual adaptation. Normative and policy efforts, such as guidelines for demonstration data collection and transparency about which tasks a continually updated policy has been validated on, may complement technical mitigation. We view this paper as a foundational empirical analysis intended to inform safer and more efficient continual VLA research rather than as a deployment recipe.

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

# Appendix: Table of Contents

# A. More Continual Learning Results

## A.1. Confusion Matrix Results for Comparison

We present the full confusion matrix results in Figure 9, which extends Figure 1: each entry $(i, j)$ denotes the success rate on task $j$ after training on checkpoint $i$ under ER, for `Pi0`, `GR00T N1.5`, `BC-Transformer`, `BC-Diffusion`, and `BC-ViT` on all four LIBERO benchmarks.

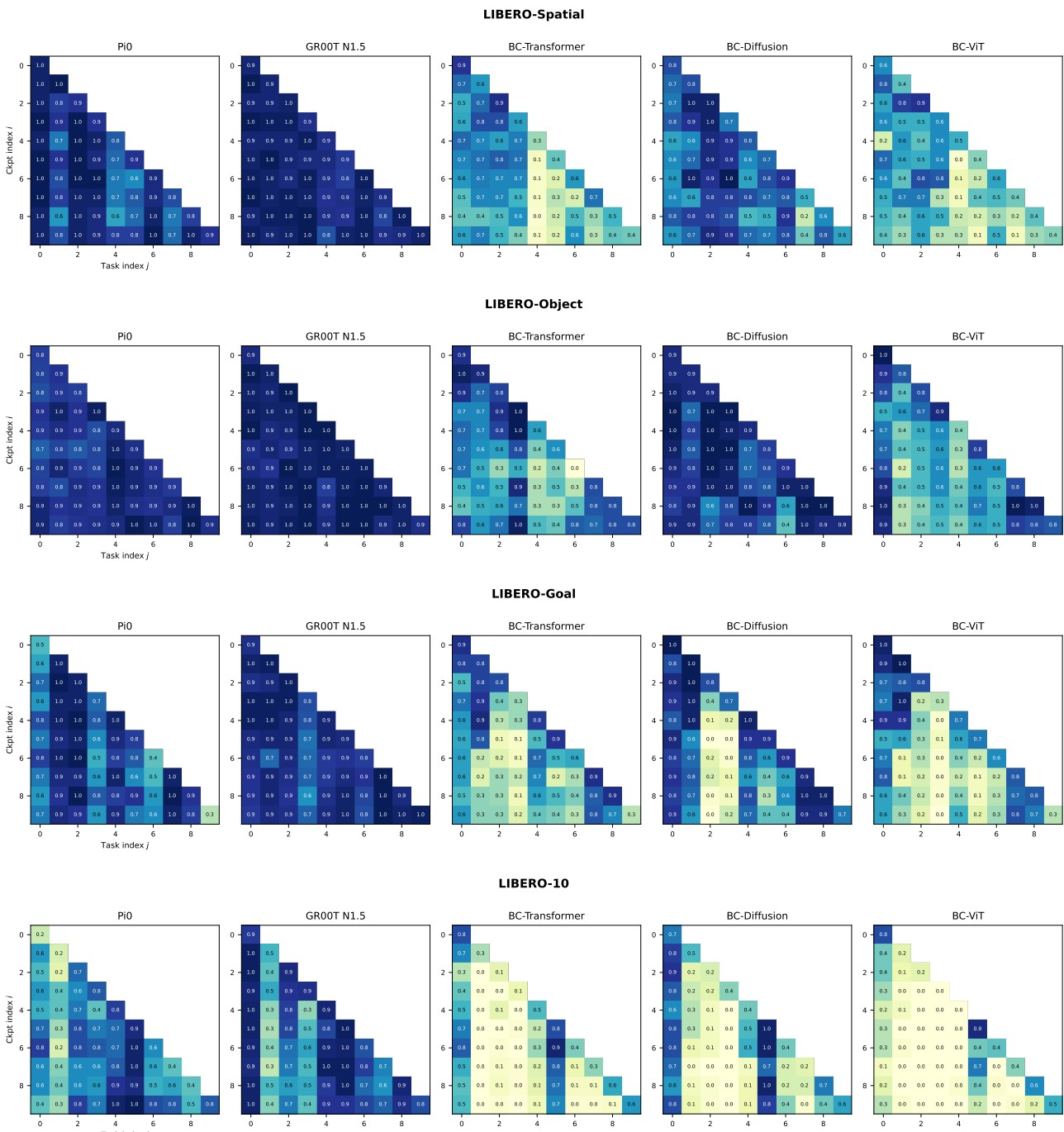

*Figure 9.* **Full confusion matrix results (extending Figure 1).** As in Figure 1, each entry $(i, j)$ denotes the success rate on Task $j$ after training on checkpoint $i$ under ER. Shown here are the complete results for `Pi0`, `GR00T N1.5`, `BC-Transformer`, `BC-Diffusion`, and `BC-ViT` on the four LIBERO benchmarks.

## A.2. LIBERO-10 Continual Learning Results

We discuss LIBERO-10 separately because, unlike the other three benchmarks, LIBERO-10 comprises diverse tasks with distinct scenes and objectives, which makes it the most challenging benchmark for the continual learning setting. The performance on this benchmark differs a lot across different methods, which reveals an interesting limitation of the standard NBT metric (see Section 3). Since this discussion concerns the metric definition itself rather than the core findings of our work, we present it here in the appendix.

**Normalized NBT.** Recall that $\text{NBT}_k = \frac{1}{K-k} \sum_{\tau=k+1}^{K} (c_{k,k} - c_{k,\tau})$, where $c_{k,k}$ is the success rate on task $k$ immediately after learning it, and $c_{k,\tau}$ is the success rate on task $k$ after training through task $\tau$. Under this definition, policies with higher initial success rates are penalized more heavily: a drop from 90% to 0% contributes 0.90 to NBT, whereas a drop from 40% to 0% contributes only 0.40, even though both represent complete forgetting. To address this, we introduce a *normalized* NBT variant that scales each forgetting term by the initial performance $c_{k,k}$:

$$\text{NBT}_k^{\text{norm}} = \frac{1}{K-k} \sum_{\tau=k+1}^{K} \frac{c_{k,k} - c_{k,\tau}}{c_{k,k}}, \tag{1}$$

so that complete forgetting always equals 1.0, regardless of the initial success rate. Tasks with $c_{k,k} = 0$ are excluded since no meaningful forgetting can be measured.

As shown in Figure 10, the absolute NBT (left) makes it appear that GR00T suffers the most forgetting at small buffer sizes because it starts from high success rates. The normalized NBT (right) reveals a more nuanced picture: all policies experience near-complete relative forgetting at buffer size = 10, while the VLA models (Pi0, GR00T) retain substantially more knowledge at buffer size = 1000, with their normalized NBT dropping to near zero or even becoming negative (indicating performance improvement on earlier tasks). We leave a more systematic study of forgetting metrics to future work.

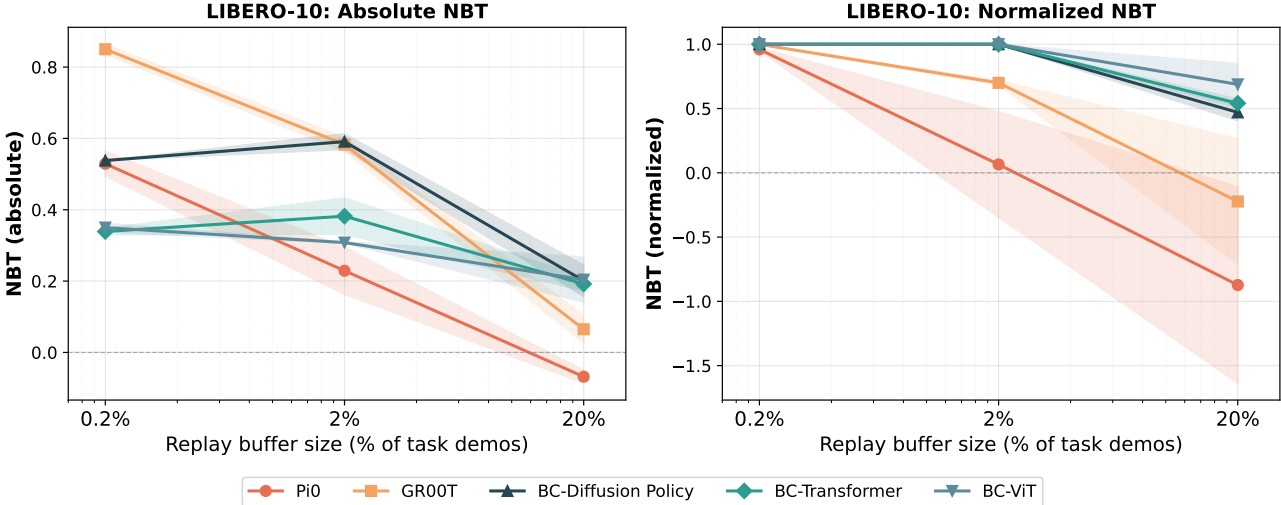

*Figure 10.* **Absolute vs. normalized NBT on LIBERO-10.** Left: standard NBT, which penalizes higher-performing policies more heavily. Right: NBT normalized by the initial (diagonal) success rate, where 1.0 indicates complete forgetting and values below zero indicate performance improvement after learning later tasks.

*Table 6.* **Continual learning performance on `LIBERO-10` for different replay buffer sample sizes**: Average Success Rate (SR) and Negative Backward Transfer (NBT) for sample sizes 10, 100, and 1000.

| Method | Sample Size = 10 | | Sample Size = 100 | | Sample Size = 1000 | |
|---|---|---|---|---|---|---|
| | SR (↑) | NBT (↓) | SR (↑) | NBT (↓) | SR (↑) | NBT (↓) |
| Pi0 | $0.576 \pm 0.034$ | $0.529 \pm 0.035$ | $0.584 \pm 0.045$ | $0.229 \pm 0.066$ | $0.563 \pm 0.028$ | $-0.068 \pm 0.018$ |
| GR00T | $0.851 \pm 0.007$ | $0.766 \pm 0.013$ | $0.854 \pm 0.011$ | $0.525 \pm 0.018$ | $0.820 \pm 0.017$ | $0.059 \pm 0.035$ |
| BC-Diffusion Policy | $0.568 \pm 0.020$ | $0.538 \pm 0.031$ | $0.616 \pm 0.012$ | $0.591 \pm 0.022$ | $0.464 \pm 0.024$ | $0.201 \pm 0.044$ |
| BC-Transformer | $0.367 \pm 0.005$ | $0.339 \pm 0.009$ | $0.403 \pm 0.058$ | $0.382 \pm 0.051$ | $0.376 \pm 0.034$ | $0.192 \pm 0.019$ |
| BC-ViT | $0.371 \pm 0.015$ | $0.350 \pm 0.012$ | $0.324 \pm 0.020$ | $0.308 \pm 0.002$ | $0.316 \pm 0.062$ | $0.204 \pm 0.063$ |

## A.3. Results for Other Replay Buffer Sizes

*Table 7.* **Continual learning performance on LIBERO benchmarks (sample size = 100)**: Average Success Rate (SR) and Negative Backward Transfer (NBT). `LIBERO-10` results in Tab. 6.

| Method | LIBERO-Spatial | | LIBERO-Object | | LIBERO-Goal | | Average | |
|---|---|---|---|---|---|---|---|---|
| | SR (↑) | NBT (↓) | SR (↑) | NBT (↓) | SR (↑) | NBT (↓) | SR (↑) | NBT (↓) |
| Pi0 | $0.899 \pm 0.012$ | $0.155 \pm 0.010$ | $0.885 \pm 0.020$ | $0.043 \pm 0.072$ | $0.729 \pm 0.016$ | $0.153 \pm 0.039$ | $0.838 \pm 0.017$ | $0.117 \pm 0.048$ |
| GR00T | $0.921 \pm 0.017$ | $0.151 \pm 0.036$ | $0.963 \pm 0.015$ | $0.117 \pm 0.044$ | $0.939 \pm 0.008$ | $0.282 \pm 0.031$ | $0.941 \pm 0.014$ | $0.184 \pm 0.037$ |
| BC-Diffusion Policy | $0.729 \pm 0.094$ | $0.414 \pm 0.117$ | $0.788 \pm 0.083$ | $0.430 \pm 0.080$ | $0.793 \pm 0.035$ | $0.714 \pm 0.059$ | $0.770 \pm 0.075$ | $0.520 \pm 0.089$ |
| BC-Transformer | $0.657 \pm 0.009$ | $0.379 \pm 0.028$ | $0.625 \pm 0.188$ | $0.402 \pm 0.125$ | $0.744 \pm 0.017$ | $0.609 \pm 0.024$ | $0.675 \pm 0.109$ | $0.463 \pm 0.076$ |
| BC-ViT | $0.537 \pm 0.048$ | $0.444 \pm 0.042$ | $0.596 \pm 0.275$ | $0.380 \pm 0.239$ | $0.693 \pm 0.015$ | $0.653 \pm 0.022$ | $0.609 \pm 0.161$ | $0.492 \pm 0.141$ |

*Table 8.* **Continual learning performance on LIBERO benchmarks (sample size = 10)**: Average Success Rate (SR) and Negative Backward Transfer (NBT). `LIBERO-10` results in Tab. 6.

| Method | LIBERO-Spatial | | LIBERO-Object | | LIBERO-Goal | | Average | |
|---|---|---|---|---|---|---|---|---|
| | SR (↑) | NBT (↓) | SR (↑) | NBT (↓) | SR (↑) | NBT (↓) | SR (↑) | NBT (↓) |
| Pi0 | $0.872 \pm 0.018$ | $0.704 \pm 0.062$ | $0.875 \pm 0.010$ | $0.511 \pm 0.050$ | $0.773 \pm 0.036$ | $0.536 \pm 0.049$ | $0.840 \pm 0.024$ | $0.583 \pm 0.054$ |
| GR00T | $0.910 \pm 0.010$ | $0.710 \pm 0.013$ | $0.963 \pm 0.008$ | $0.750 \pm 0.017$ | $0.951 \pm 0.008$ | $0.780 \pm 0.021$ | $0.942 \pm 0.009$ | $0.747 \pm 0.018$ |
| BC-Diffusion Policy | $0.753 \pm 0.040$ | $0.779 \pm 0.049$ | $0.890 \pm 0.024$ | $0.697 \pm 0.057$ | $0.876 \pm 0.026$ | $0.865 \pm 0.022$ | $0.839 \pm 0.031$ | $0.780 \pm 0.045$ |
| BC-Transformer | $0.720 \pm 0.011$ | $0.740 \pm 0.022$ | $0.631 \pm 0.081$ | $0.536 \pm 0.045$ | $0.845 \pm 0.020$ | $0.853 \pm 0.016$ | $0.732 \pm 0.037$ | $0.710 \pm 0.028$ |
| BC-ViT | $0.609 \pm 0.056$ | $0.617 \pm 0.063$ | $0.585 \pm 0.146$ | $0.371 \pm 0.224$ | $0.753 \pm 0.030$ | $0.803 \pm 0.031$ | $0.649 \pm 0.092$ | $0.597 \pm 0.136$ |

## B. Experimental Setup Details

### B.1. ER Training Details

We fixed the replay buffer size budget for all methods to ensure fair comparison. Following the same setting as LIBERO (Liu et al., 2023b), we use a replay buffer size = 1000 transitions (*i.e.*, state-action pairs) per task across all methods, which is approximately $15 - 20\%$ of the full task dataset. Following `libero` (Liu et al., 2023b), we equally sample data from the replay buffer and the current task, making it a 1:1 ratio between the current task and all past tasks.

### B.2. LIBERO Benchmark Details

We provide the task order used for each LIBERO benchmark below. We follow the same randomized task order for all methods to ensure fair comparison.

**LIBERO-Spatial Task Order**

1. Pick up the black bowl between the plate and the ramekin and place it on the plate.
2. Pick up the black bowl next to the ramekin and place it on the plate.
3. Pick up the black bowl from table center and place it on the plate.
4. Pick up the black bowl on the cookie box and place it on the plate.
5. Pick up the black bowl in the top drawer of the wooden cabinet and place it on the plate.
6. Pick up the black bowl on the ramekin and place it on the plate.
7. Pick up the black bowl next to the cookie box and place it on the plate.
8. Pick up the black bowl on the stove and place it on the plate.
9. Pick up the black bowl next to the plate and place it on the plate.
10. Pick up the black bowl on the wooden cabinet and place it on the plate.

**LIBERO-Object Task Order**

1. Pick up the alphabet soup and place it in the basket.
2. Pick up the bbq sauce and place it in the basket.
3. Pick up the butter and place it in the basket.
4. Pick up the chocolate pudding and place it in the basket.
5. Pick up the cream cheese and place it in the basket.
6. Pick up the ketchup and place it in the basket.
7. Pick up the milk and place it in the basket.
8. Pick up the orange juice and place it in the basket.
9. Pick up the salad dressing and place it in the basket.
10. Pick up the tomato sauce and place it in the basket.

**LIBERO-Goal Task Order**

1. Open the middle drawer of the cabinet.
2. Put the bowl on the stove.
3. Put the wine bottle on top of the cabinet.
4. Open the top drawer and put the bowl inside.
5. Put the bowl on top of the cabinet.
6. Push the plate to the front of the stove.
7. Put the cream cheese in the bowl.
8. Turn on the stove.
9. Put the bowl on the plate.
10. Put the wine bottle on the rack.

---

**LIBERO-10 Task Order**

1. Pick up the book and place it in the back compartment of the caddy.

2. Put both moka pots on the stove.

3. Put both the alphabet soup and the cream cheese box in the basket.

4. Put both the alphabet soup and the tomato sauce in the basket.

5. Put both the cream cheese box and the butter in the basket.

6. Put the black bowl in the bottom drawer of the cabinet and close it.

7. Put the white mug on the left plate and put the yellow and white mug on the right plate.

8. Put the white mug on the plate and put the chocolate pudding to the right of the plate.

9. Put the yellow and white mug in the microwave and close it.

10. Turn on the stove and put the moka pot on it.

---

## B.3. Model Details

*Table 9.* Architecture and pretraining details of all models evaluated in this work. "Frozen" indicates components whose weights are not updated during continual learning; "Finetuned" indicates trainable components. For Pi0, LoRA adapters are applied to the frozen backbone.

| | pi0 | GR00T | BC-Diffusion Policy | BC-Transformer | BC-ViT |
|---|---|---|---|---|---|
| Total params | 3B | 3B | ∼26M | ∼15M | ∼15M |
| Vision encoder | SigLIP-So400m (pretrained, finetuned) | SigLIP (pretrained, frozen) | ResNet-18 (trained from scratch) | ResNet-18 (trained from scratch) | ViT-Patch (trained from scratch) |
| Language model | Gemma-2B (pretrained, finetuned) | Qwen3-1.7B (pretrained, frozen) | BERT (pretrained, frozen) + MLP (trained from scratch) | BERT (pretrained, frozen) + MLP (trained from scratch) | BERT (pretrained, frozen) + MLP (trained from scratch) |
| Action head | Flow Matching (Gemma-300M) (pretrained, finetuned) | Flow Matching DiT (pretrained, finetuned) | DDPM UNet (trained from scratch) | MLP + GMM (trained from scratch) | MLP + GMM (trained from scratch) |
| Finetuning strategy | Vision: full FT, Language: LoRA FT, Action: LoRA FT | Vision: frozen, Language: frozen, action: full FT | Full FT (all params) | Full FT (all params) | Full FT (all params) |

## B.4. Training Hyperparameters

*Table 10.* Training hyperparameters for continual learning experiments. VLA models (pi0, GR00T) use a fixed number of gradient steps per task, while BC baselines train for a fixed number of epochs per task.

| Hyperparameter | pi0 | GR00T | BC-Diffusion Policy | BC-Transformer | BC-ViT |
|---|---|---|---|---|---|
| Learning rate | 2.5e-5 | 1e-4 | 1e-4 | 1e-4 | 1e-4 |
| Batch size | 8 | 16 | 16 | 16 | 16 |
| Training budget per task | 10,000 steps | 10,000 steps | 50 epochs | 50 epochs | 50 epochs |
| Optimizer | AdamW | AdamW | AdamW | AdamW | AdamW |
| Weight decay | 1e-10 | 1e-5 | 1e-4 | 1e-4 | 1e-4 |
| LR scheduler | Cosine decay | Cosine decay | Cosine decay | Cosine decay | Cosine decay |

## B.5. Continual Learning Baselines

**Elastic Weight Consolidation (EWC)** (Kirkpatrick et al., 2017b) is another common continual learning method that employs a regularization term discouraging changes to parameters important for previously learned tasks. EWC estimates parameter importance using Fisher information matrix computed after each task, and uses it to weight a quadratic penalty that keeps the model close to the parameters learned from earlier tasks.

**Sequential Training** (Liu et al., 2023b) is the simplest continual learning method, where the model is trained on each task sequentially without any explicit mechanisms to prevent forgetting.

## C. Study on Other Factors that Contribute to VLA's Continual Learning Behavior

**Model size contributes to mitigating forgetting.** We observe that large VLA model trained from scratch (without pretraining) also displays near-zero forgetting; one hypothesis is model size also contributes to VLA's forgetting dynamics. To validate on the effect of model size, we test on a few Pi0 variants with smaller model sizes (all trained from scratch). We observe in Tab. 11 that smaller models gives higher NBT than larger ones, showing that model size also plays a role here. While this serves an initial investigation, we will leave more systematic study of model size (in relation to pretraining) in future work.

*Table 11.* **Effect of model size on forgetting (scratch training).** NBT (positive value indicates there is forgetting; the lower is better) for Pi0 variants trained from scratch with different LLM-action expert and vision-backbone sizes.

| LLM + Action Expert | Vision Backbone | NBT ($\downarrow$) |
|---|---|---|
| 17M | ResNet ($\sim$0.5M) | 0.1100 |
| 17M | SigLIP-B/16 (80M) | 0.0628 |
| 17M | SigLIP-So400M/14 (400M) | 0.0264 |
| 250M | SigLIP-B/16 (80M) | -0.0478 |
| 250M | SigLIP-So400M/14 (400M) | -0.0520 |

**Training objective has little effect on forgetting.** To isolate the effect of the training objective, we ablate Pi0, which trains its action expert with a flow-matching objective, by retraining it with an $\ell_2$ regression loss while keeping the architecture unchanged. As shown in Tab. 12, the resulting performance has little difference, suggesting that the choice of training objective is not a primary factor.

*Table 12.* Comparison of continual learning metrics for different training objectives (LIBERO-Spatial).

| Method | SR | NBT |
|---|---|---|
| Pi0 w. Flow Matching | 0.836 | -0.0003 |
| Pi0 w. L2 Loss | 0.853 | 0.016 |

## D. Per-Task Knowledge Transfer Curves: Pi0 vs. BC-Transformer

Extending Figure 8, we present per-task breakdowns of the knowledge transfer curves comparing Pi0 and BC-Transformer across all four LIBERO benchmarks (See Figure 11, 12, 13, and 14). For each task, the top subplot shows Pi0 and the bottom shows BC-Transformer, with both the Finetuning and Learn First Time (LFT) curves plotted. All success rates are normalized by the LFT curve's peak for that task; a value of 1.0 indicates full recovery of LFT performance.

**LIBERO-Spatial**

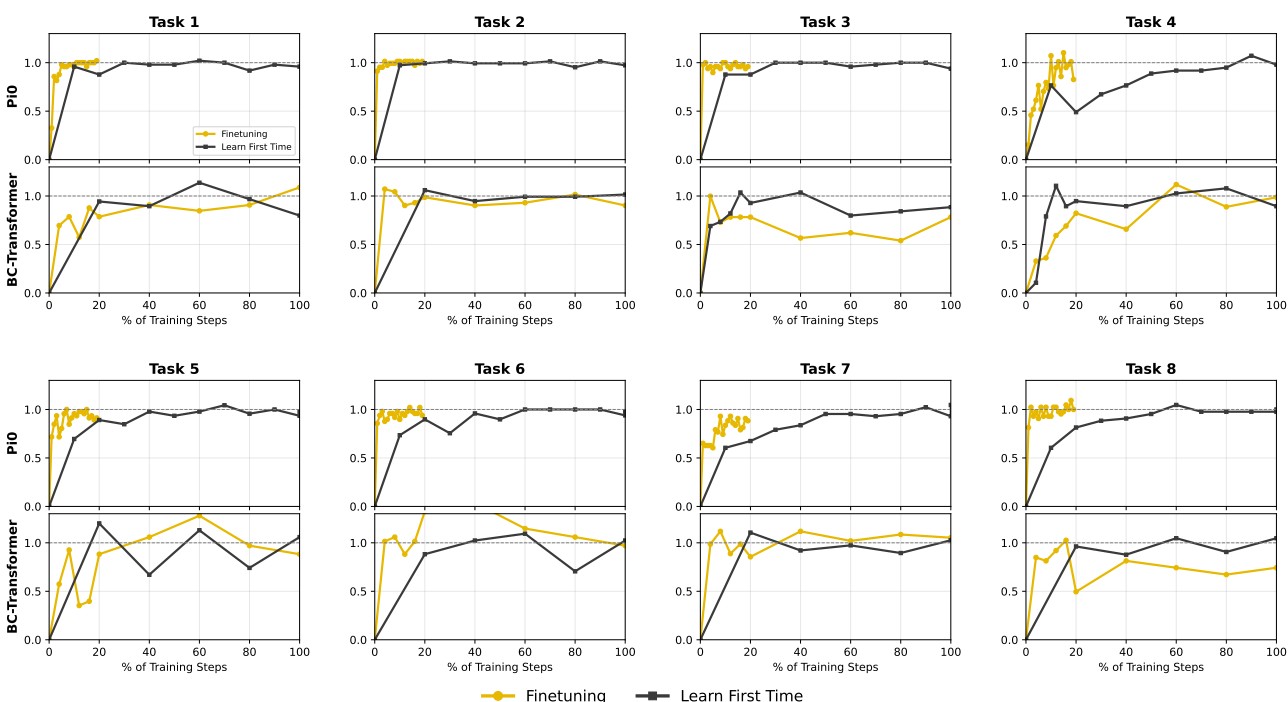

*Figure 11.* **Per-task knowledge transfer curves on** `LIBERO-Spatial`.

**LIBERO-Object**

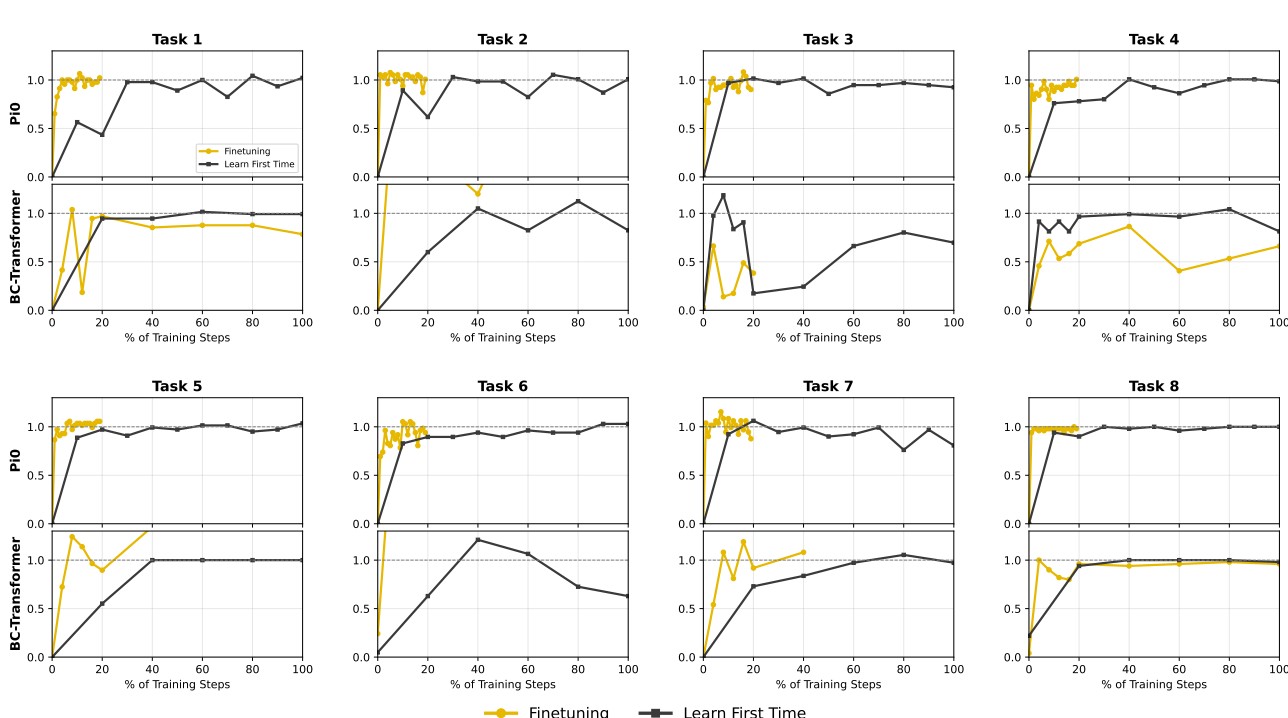

*Figure 12.* **Per-task knowledge transfer curves on** `LIBERO-Object`.

**LIBERO-Goal**

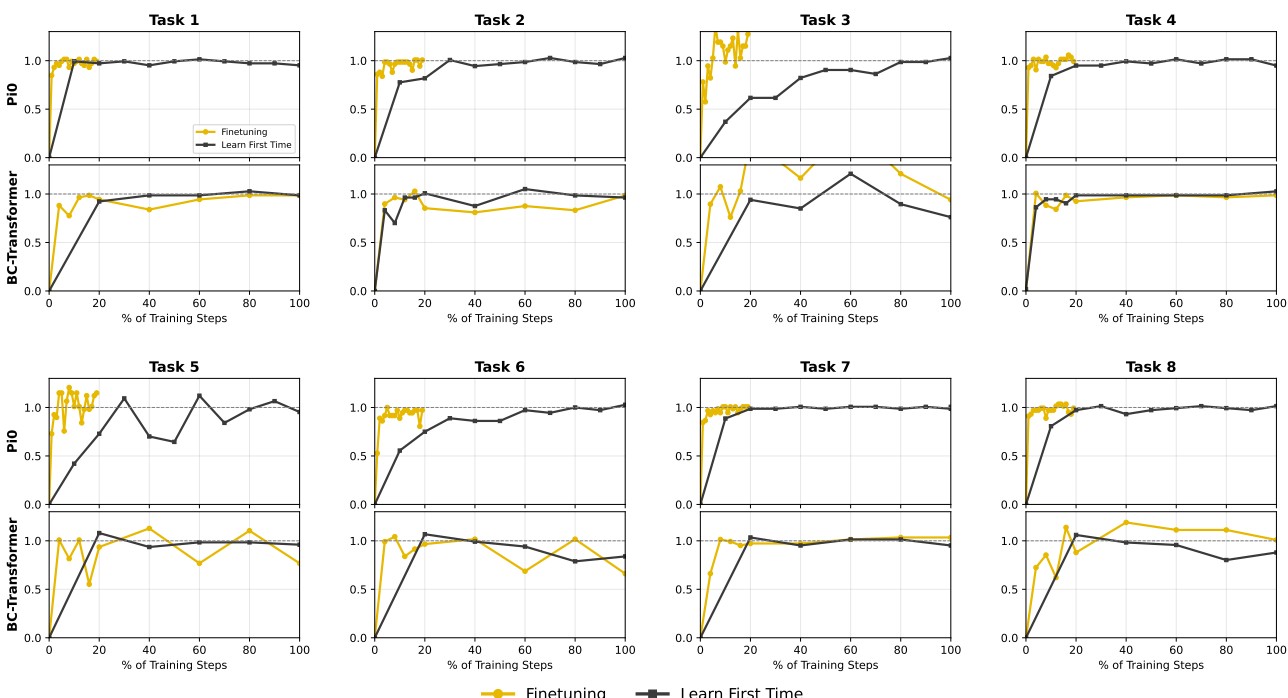

*Figure 13.* **Per-task knowledge transfer curves on `LIBERO-Goal`.**

**LIBERO-10**

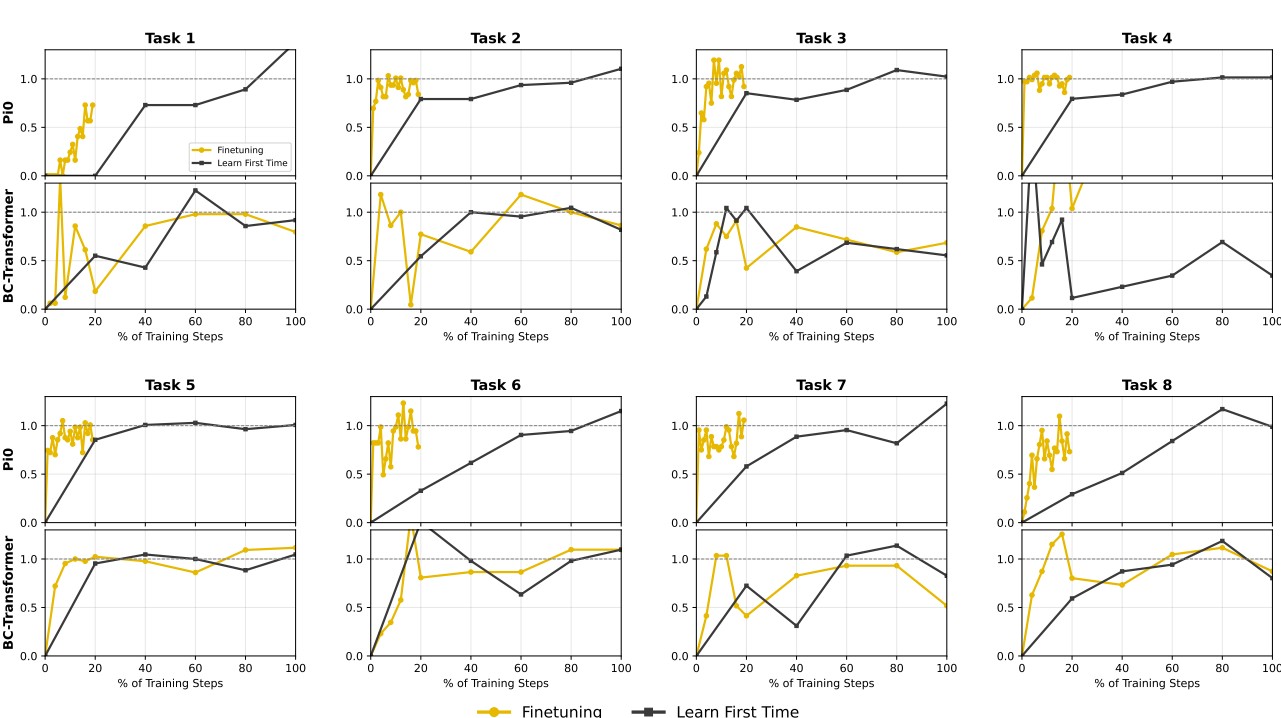

*Figure 14.* **Per-task knowledge transfer curves on `LIBERO-10`.**

