# OpenReview forum: "Pretrained Vision-Language-Action Models are Surprisingly Resistant to Forgetting in Continual Learning"
_ICML.cc/2026/Conference — ICML 2026 spotlight_

### Official Review · Reviewer_A6Tv · 2026-02-26

**Soundness:** 3
**Presentation:** 4
**Significance:** 2
**Originality:** 2
**Overall Recommendation:** 4
**Confidence:** 4

**Summary:**

This paper empirically studies knowledge retention under sequentially learning task in a robotics environment. It compares large-scale pre-trained Vision-Language Action (VLA) models with a non-pretrained small baseline model. As a comparison dataset, different suites from LIBERO are used. Using different buffer capacities, the authors show that pre-trained model are robust to forgetting. Using only minimal past-task samples, pre-trained models can maintain near-perfect performance even on historic tasks. This is contrasted with the from-scratch model (that lacks prior pretraining knowledge) where forgetting is prevalent. Several finegrained analyses are conducted by the authors to study the occurence of (non-)forgetting.

**Compliance With Llm Reviewing Policy:**

Affirmed.

**Final Justification:**

My final recommendation is a weak accept. The rebuttal did not change my assessment of the paper.
For my recommendation, I weighted the studied setting (CL + VLA + forgetting) with its orginality and significance.
While I appreciate the experiments done, also in response to the other reviewers, I think the paper is technically solid, but I do not see a sufficient novelty to further raise my score.

**Key Questions For Authors:**

1. Figure 7: How can the success rate percentage jump over 100%? (E.g., second column, second row, blue line)
2. Section 2.2. You mention that VLAs need an observation history? How is this related to the replay buffer? I presume it occupies a different buffer?
3. (Suggestion) Section 1, lines 100 and 057: replace found with find to align the tenses.
4. (Suggestion): Place positioning (currently on page 8) to the front (maybe along with the entire related work)
5. Could it be that the data from LIBERO is not sufficient enough a-priori to effectively train a model? I take this from Table 3, where no-pretrained models essentially have similar performances. This suggest that the chosen architectures are overkill for this problem and underfit. (In line, I assume that LIBERO is not part of any model's pretraining data).

**Limitations:**

Please include an implications statement. Replaying data might not always be possible due to privacy (real-world robot env.) or other regulations.

**Strengths And Weaknesses:**

The paper uses a clear logical structure that allowed easily following the authors' arguments. The general presentation is clean, thought the ordering of Figures and Tables needs to be improved: often, the described Figure/Table is on the prior or next page, necessitating jumping around.

The paper is technically sound and claims are well supported by the usage of three differently pretrained models, and I appreciate the experimental efforts required. However, I fail to appreciate the claims' significance and novelty: Across many domains, pre-training has been shown to improve the performance, presumably due to large-scale knowledge encoded in the parameters. It is thus not that surpising that information are to some extend robust to forgetting, simply because other knowledge could make up for the loss. (I am open for discussion on this point.)

---

> ### Author Rebuttal · Authors · 2026-03-31
>
> We thank the reviewer for the thoughtful review and for recognizing our paper's clear logical structure, excellent presentation, and experimental rigor across three distinct pretrained models. We address each point below.
>
> **Q1: Significance and novelty**
>
> We agree that pretraining often helps downstream performance. Our contribution is more specific: under standard robotic CL benchmarks, simple ER with VLAs yields qualitatively different forgetting dynamics than classical small BC policies. These include near-zero or negative NBT at small replay fractions (as low as 2% of the training data), plus recovery evidence suggesting representational retention — phenomena not explained by "higher accuracy implies lower NBT" alone. In particular:
>
> - The Pareto frontier analysis (Fig. 5) shows the gap between pretrained and non-pretrained models widens as the replay buffer shrinks, revealing a data-efficiency regime unique to pretrained VLAs.
>
> - The knowledge retention experiments (component-swapping and recovery) (Sec. 5) demonstrate that even when task-level performance degrades, VLAs retain knowledge in their internal representations — a finding that goes beyond the general observation that pretraining helps performance.
>
> We will sharpen the novelty framing in the revision to emphasize these empirical dynamics and controlled ablations.
>
> **Q2: Figure 7 — success rate percentage jumping over 100%.**
>
> Thanks for this observation! We note that for the fine-tuned version, the success rate percentage is a ratio relative to the best success rate learned for the first time; this ratio can exceed 100% if the success rate is very high. The best success rate of learned first time is the average of the K top success rates, and therefore, the individual number could be higher than 100%.
>
> **Q3: Observation history vs. replay buffer.**
>
> Observation history refers to inputs within a rollout as implemented by the VLA; the replay buffer stores past-task demonstration transitions for training.
>
> **Q4: Tense suggestion (lines 100 and 057).**
>
> Thanks for spotting this! We will unify the tense in the opening passage ("found" → "find") for consistency.
>
> **Q5: Positioning of related work.**
>
> We will strengthen early positioning and consider moving Related Work earlier (currently on page 8) to better orient readers before the experimental sections.
>
> **Q6: Is LIBERO data insufficient to effectively train a model? No-pretrained models essentially have similar performances.**
>
> Thanks for bringing this important point! We would like to note that although the two models display similar SRs, this could imply fundamentally different CL dynamics. For Pi0, the success rate is due to optimization issue of training large models on small scale data; for BC-Transformer, it is a problem of underfitting - using more advanced, similarly non-pretrained architectures like Diffusion Policy gives higher success rate (10% higher). Furthermore, while Pi0 from scratch and BC-Transformer achieve comparable average success rates in Table 3 (0.655 vs. 0.678), their forgetting behaviors are qualitatively different: Pi0 from scratch achieves NBT of −0.039 (slight positive backward transfer), whereas BC-Transformer suffers NBT of 0.191 (substantial forgetting).
>
> **Q7: Limitations**
>
> We will add a limitations section in the revised paper:
> - Replay buffer practicality and privacy concerns: Experience replay assumes stored transitions from past tasks, which may face privacy or storage constraints in real-world deployments.
> - Sim-to-real gap: All experiments are conducted in simulation; real-world validation with sensor noise and physical dynamics are needed to confirm quantitative findings.
> - Benchmark scope: LIBERO tasks share the same embodiment and similar environments.

---

> > ### Author Rebuttal · Reviewer_A6Tv · 2026-04-01
> >
> > I thank the authors for providing a rebuttal to my review. I trust them to integrate the changes (description of Fig 7, novelty framing, related work positioning, discussion of the LIBERO data, expanded limitation/implications section) into the paper.
> >
> > Given that my score is already positive, I will maintain it.

---

### Official Review · Reviewer_TB6V · 2026-03-09

**Soundness:** 1
**Presentation:** 2
**Significance:** 3
**Originality:** 2
**Overall Recommendation:** 4
**Confidence:** 2

**Summary:**

This paper is devoted to an empirical study of continual learning in the context of Visual-Language-Action (VLA) models. To enhance continual learning capabilities, the authors add a small Experience Replay (ER) buffer to the dataset on which the VLA is trained to solve a new task in a sequential supervised fine-tuning (SFT) setting (the ER includes subsets of datasets for other prior tasks). Experimental results with several popular VLA models on the LIBERO benchmark show that large-scale pretraining improves continual learning compared to training from scratch baselines, and that using a simple ER buffer yields metrics comparable to or better than other continual learning baselines.

**Compliance With Llm Reviewing Policy:**

Affirmed.

**Final Justification:**

The new results answer most of my questions, and I am changing my assessment from 2 to 4, with the caveat that all new experiments should be included in the camera-ready version with a more detailed analysis.

**Key Questions For Authors:**

Overall, the work is interesting and covers a very important area of ​​VLA training in robotics. However, the work has a number of problems, both in terms of presentation and positioning, and in terms of methodology. I'm open to discussion and am prepared to reconsider my scores if the authors can address my comments and questions.

**Limitations:**

Limitations section is not included in the text. At this point, it is unknown how the proposed method will perform during continuous learning on different embodiments, different benchmarks, and also on real robots.

**Strengths And Weaknesses:**

**Strengths:**

1. This work addresses an important and pressing problem, as training a VLA for a new task is an expensive procedure, and we would like to ensure that during this procedure the model's performance at least does not drop on tasks it already knows how to solve.

2. A large number of tables and figures reflecting experimental results

3. A simple and clear idea of ​​using ER


**Weaknesses and Questions:**
1. The emphasis in the paper's narrative creates a misunderstanding of the paper's essence upon first reading. For example, the most significant outcome after reading is that adding ER to sequential SFT for VLAs enables successful performance on continual learning tasks. However, the visual abstract and the beginning of the text create the impression that the paper will primarily study the effect of large-scale pretraining on this ability. For example, wouldn't it be better to change GR00T vs. BC-Transformer in the visual abstract (Figure 1) to GR00T+ER vs. GR00T to specifically highlight the effect of ER? I also recommend rewriting the central question (L89-91) ("Do large pretrained VLAs behave differently in continual learning from smaller models, and if so, in what ways?"), since the actual object of study in the paper is broader.

2. The phrase "acquire new skills over time" appears several times at the beginning of the text, making it unclear whether this refers to online continuous learning or SFT. I think it's better to clearly state from the start that the SFT approach is being discussed throughout the paper.

3. Section 2.1, H - episode timeout? Why should it be the same for all tasks?

4. The formula for the objective (L096, as well as L147) – why is $o_{\leq t}$ there? This is incorrect, since, as far as I know, the VLAs discussed in the paper (GR00T, pi0, OpenVLA-OFT) do not process observations from previous steps and only operate with information from the current timestep. This expression needs to be rewritten more correctly (the sum should also include summation up to $l_p-1$).

5. The paper text is quite difficult to read, as the tables and graphs associated with it are located 1-2 pages above the text, forcing constant scrolling. I suspect this problem arises because the introduction is too short, causing this offset. Perhaps this could be resolved by moving the "Related Work" section to the beginning of the paper.

6. Despite the fact that LIBERO is a very popular benchmark, it would be worthwhile to describe in more detail (at least in the appendix) what kind of benchmark it is, how its suites differ, how the tasks in the suites differ: how similar they are, what changes in them, how much data is included, etc.

7. The work lacks information about the parameters of the models and their training.

8. L139 - sampled transitions per task or trajectories? Also, how should we interpret replay buffer size = 10 (Figure 4) if, for example, LIBERO-Object only contains 10 tasks?

9. L146 (for each $k \in [K]$) - $K$ is a natural number, so this expression is unclear to me.

10. The paper doesn't check the statistical significance of the results anywhere, which is critical. For example, in addition to the average values ​​in the tables, it would be nice to see the standard error of the mean for several seeds. Furthermore, the tables don't make it clear how to interpret the scale of the NBP metrics: for example, are "0.0009" and "-0.00013" a large or a small value? Are the 4th or 5th decimal places important, or can they be considered zero?

11. I was not convinced by the evidence supporting the claim "Pretraining Plays an Integral Role in Improving Continual Learning Performance." Thus,

    1) To me, the results for Pi0 from VL + Action and Pi0 from VL in Table 3 look roughly the same. Moreover, pre-training on robotics data even yields a slightly worse SR.

    2) At the beginning of this paper, the authors contrasted continuous learning with multi-task learning. Couldn't we say that VLAs are trained in multi-task mode by default, meaning they are inherently more robust to new data than, say, BC-Transformer? Perhaps this isn't a matter of large-scale pretraining, but rather multi-task pretraining?

    3) It seems that the methodology shown in Figure 5 is only applicable if the objectives are similar across tasks, and the embodiments are the same. What happens if, for example, the robots change between tasks? Or if benchmarks changes between tasks?

    4) VLA is used to solve an instruction-conditioned problem. In this case, can the instruction be considered the key to the corresponding representations in the model? (The point is that the specific nature of VLA's work with instructions helps the model separate representations during retraining for new tasks.) To answer this question, the authors need to conduct an experiment where the same text instruction is used for all tasks or not at all (where the task is defined at the observation level), and conduct similar experiments with retraining.

12. How does the distribution within the ER change as new data is added during continuous learning? It seems that in LIBERO, the data is highly correlated between tasks, and there won't be any significant shifts. What happens if you first train on LIBERO and then on a different benchmark?

13. (minor) It would be great to see a diagram showing how continual learning occurs using ER, and how it changes as learning progresses.

---

> ### Author Rebuttal · Authors · 2026-03-31
>
> We thank the reviewer for the detailed feedback and openness to discussion. We address all points below.
>
> **Narrative emphasis and Figure 1.** Both GR00T and BC-Transformer in Figure 1 use the *same* sequential SFT + ER protocol; the teaser isolates *pretrained VLA vs. non-pretrained BC*, rather than "ER vs. no ER." We will make this explicit, add an appendix panel contrasting ER vs. no-replay on the same backbone, and rewrite the central research question (L89–91) to reflect the paper's full scope (including replay, pretraining ablations, and mechanistic analyses).
>
> **"Acquire new skills over time."** We will clarify upfront that this refers to sequential offline SFT over a task sequence, not online RL.
>
> **Float placement.** We will rebalance intro length and use `\FloatBarrier` so figures sit nearer their first reference, and consider moving Related Work earlier.
>
> **Shared horizon H (Sec 2.1).** This follows the standard LIBERO MDP abstraction; tasks share dynamics/horizon but differ in initial states and goals. Episodes terminate at $l_k \le H$. We will add a clarifying sentence.
>
> **BC objective (L096, L147).** Corrected: upper limit changed to $l_p{-}1$, observation symbol to $o_t$, removing misleading $\pi(o_{\le t})$ notation.
>
> **L139 — transitions vs. trajectories.** Buffer size = M means M stored *(state, action) transitions per past task*, not trajectories and not tied to the number of tasks. "Buffer size = 10" means 10 transitions per previous task. We will clarify in text and Figure 4's caption.
>
> **L146 notation.** Will replace with $k \in \{1,\ldots,K\}$ following Liu et al.
>
> **LIBERO description.** We will add a suite-by-suite appendix description: Spatial (varying object placements), Object (varying object instances, shared goals), Goal (diverse multi-step interactions), LIBERO-10 (long-horizon, visually heterogeneous—the hardest suite).
>
> **Model/training details.** We will add appendix tables with architecture sizes (Pi0: 3B, GR00T: 3B, BC baselines: 15–26M), frozen vs. finetuned components, and hyperparameters (LR, batch size, steps/epochs per task, optimizer, scheduler) for all five models, with a forward pointer from the main text.
>
> **Statistical significance and NBT interpretation.** We have updated all main results with mean ± std over 3 seeds. Key updated numbers (Avg SR / Avg NBT): Pi0 **0.768±0.017 / −0.016±0.022**; GR00T **0.919±0.011 / 0.027±0.021**; BC-Transformer **0.585±0.066 / 0.245±0.080**. NBT is on the same [0,1] scale as SR; values near $10^{-3}$–$10^{-4}$ indicate negligible backward transfer, not sub-percent precision claims. We will add an explicit interpretation note.
>
> **Pretraining claim (Table 3).** The difference between Pi0 VL+Action and Pi0 VL manifests as a *trade-off* between forward SR and forgetting, with robotics pretraining mattering most under *small replay* (Pareto frontier, Fig. 3). At large replay (1000), both variants perform similarly since sufficient replay already mitigates forgetting.
>
> **Multi-task vs. large-scale pretraining.** VLAs are *not* trained on LIBERO in multi-task joint mode; their pretraining comes from diverse internet-scale vision-language corpora and/or different-domain robot data—fundamentally different from multi-task training on downstream tasks. BC-Transformer already uses pretrained BERT, so the gap is not "any pretraining vs. none." We will clarify this distinction.
>
> **Figure 5 methodology: different embodiments/benchmarks.** Component swapping requires shared architecture but not identical embodiments. With larger domain shifts, we expect the VL backbone to undergo bigger representational changes. We plan to include a cross-benchmark CL experiment (Pi0: LIBERO in simulation → real robot) and will report results in the camera-ready revision.
>
> **Language instructions as task keys.** We ablated text conditioning on LIBERO-10 under the same ER protocol. With task-specific instructions: **SR 0.586, NBT −0.068**. With a fixed generic instruction (e.g. just "Perform the task" for all tasks): **SR 0.530, NBT −0.111**. This shows continual learning behavior does not collapse when instructions are uninformative: NBT remains negative in both conditions. The ~6pp SR drop is consistent with language acting as a useful but *non-exclusive* task cue; task identity is distributed across modalities (vision, proprioception, language), not carried solely by the instruction.
>
> **ER diagram and limitations.** We will add (i) a diagram of the ER continual learning pipeline and (ii) a limitations subsection covering embodiment shifts, cross-benchmark generalization, sim-to-real gap, and replay constraints (storage/privacy).

---

> > ### Author Rebuttal · Reviewer_TB6V · 2026-04-02
> >
> > Thanks to the authors for their response and for the informative experiment with "Language instructions as task key". They promised to fix the presentation problems later and said they would correct several notational inaccuracies. Unfortunately, I didn't see a table with the parameters and properties of the architectures considered.
> >
> > How do you calculate mean ± std over 3 seeds? Why do you now write that "BT is on the same \[0,1\] scale as SR" if (Avg SR / Avg NBT): Pi0 0.768±0.017 / -0.016±0.022 has a negative NBP value?
> >
> > The authors' response on the topic "Multi-task vs. large-scale pretraining" didn't convince me, since the corresponding ablations weren't performed and their results weren't provided. By pretraining, I mean large-scale (internet) pretraining followed by multi-task fine-tuning on robot datasets (which is typical for all VLAs). The answer that "BC-Transformer already uses pretrained BERT" doesn't answer my comment. For example, if we fine-tune BC-Transformer in multi-task mode on the OXE dataset, how will its results change in the context of continuous learning capabilities? I'm still convinced that multi-task pretraining is crucial to unlocking these capabilities.
> >
> > "Different embodiments/benchmarks" - I don't expect the authors to conduct experiments on a real robot. My question was related to what would happen if, instead of sequential SFT on the LIBERO benchmark, we at some point perform SFT on a different benchmark with different semantics and/or a different embodiment? I still haven't found an answer to this question (at least in preliminary experiments).
> >
> > I am once again grateful to the authors for the response provided and hope to receive answers to the questions above in further discussion.

---

> > > ### Author Response · Authors · 2026-04-08
> > >
> > > We thank the reviewer for additional discussions and clarifications! These are really helpful. Here are our responses:
> > >
> > > **1. Architecture/parameter table.** We omitted this from the rebuttal text due to character constraints. We include it here:
> > >
> > > Model details
> > > | | **Pi0** | **GR00T** | **BC-Diffusion Policy** | **BC-Transformer** | **BC-ViT** |
> > > |---|:---:|:---:|:---:|:---:|:---:|
> > > | **Total params** | 3B | 3B | ~26M | ~15M | ~15M |
> > > | **Vision encoder** | SigLIP-So400m (pretrained, finetuned) | SigLIP (pretrained, frozen) | ResNet-18 (scratch) | ResNet-18 (scratch) | ViT-Patch (scratch) |
> > > | **Language model** | Gemma-2B (pretrained, finetuned) | Qwen3-1.7B (pretrained, frozen) | BERT (pretrained, frozen) + MLP (scratch) | BERT (pretrained, frozen) + MLP (scratch) | BERT (pretrained, frozen) + MLP (scratch) |
> > > | **Action head** | Flow matching / Gemma-300M (pretrained, finetuned) | Flow matching DiT (pretrained, finetuned) | DDPM UNet (scratch) | MLP + GMM (scratch) | MLP + GMM (scratch) |
> > > | **Finetuning (CL)** | Vision: full FT; Lang: LoRA; Action: LoRA | Vision: frozen; Lang: frozen; Action: full FT | Full FT | Full FT | Full FT |
> > >
> > > Training hyperparameters
> > > | **Hyperparameter** | **Pi0** | **GR00T** | **BC-Diffusion** | **BC-Transformer** | **BC-ViT** |
> > > |---|:---:|:---:|:---:|:---:|:---:|
> > > | Learning rate | 2.5e-5 | 1e-4 | 1e-4 | 1e-4 | 1e-4 |
> > > | Batch size | 8 | 16 | 16 | 16 | 16 |
> > > | Training budget/task | 10,000 steps | 10,000 steps | 50 epochs | 50 epochs | 50 epochs |
> > > | Optimizer | AdamW | AdamW | AdamW | AdamW | AdamW |
> > > | Weight decay | 1e-10 | 1e-5 | 1e-4 | 1e-4 | 1e-4 |
> > > | LR scheduler | Cosine decay | Cosine decay | Cosine decay | Cosine decay | Cosine decay |
> > > * Note: for "Training budget/task", all models are trained until convergence, following their default setup.
> > >
> > > **2. Re. mean ± std over 3 seeds and scale of NBT:**
> > >
> > > We calculate the mean and std by averaging the results across 3 seeds and taking the std of the 3 runs. To clarify, the *absolute* scale of NBT is between [0,1] (on the same scale as the success rate); the actual value of NBT is between [-1, 1].
> > >
> > > **3. Re. Multi-task vs. large-scale pretraining:**
> > >
> > > We thank the reviewer for this insightful suggestion and agree that multi-task robot pretraining is an important contributor to continual learning performance. To test this hypothesis directly, we ran an experiment using the BC-Transformer pretrained on OXE data (with Octo checkpoint: https://github.com/octo-models/octo), as the reviewer suggested. Here are the results of LIBERO-Spatial and LIBERO-Goal:
> > >
> > > | Method | Spatial SR ↑ | Spatial NBT ↓ | Goal SR ↑ | Goal NBT ↓ |
> > > |---|---:|---:|---:|---:|
> > > | Pi0 | 0.879 | 0.019 | 0.732 | -0.005 |
> > > | GR00T N1.5 | 0.940 | 0.007 | 0.943 | 0.023 |
> > > | BC-Transformer | 0.659 | 0.299 | 0.709 | 0.356 |
> > > | **BC-Transformer + OXE** | **0.340** | **0.056** | **0.349** | **0.067** |
> > >
> > > As shown below, this substantially reduces forgetting relative to the original BC-Transformer: on LIBERO-Spatial, NBT decreases from 0.299 to 0.056, and on LIBERO-Goal, NBT decreases from 0.356 to 0.067. This supports the reviewer’s hypothesis that multi-task pretraining plays an important role in improving stability.
> > >
> > > At the same time, although BC-Transformer + OXE has much less forgetting than the original BC-Transformer, its *forward transfer (success rate, SR)* is still substantially lower than that of VLAs (0.340 vs. 0.879/0.940 on LIBERO-Spatial; 0.349 vs. 0.732/0.943 on LIBERO-Goal). This suggests that *while multi-task robot pretraining improves stability, it may come at the expense of substantial plasticity*. In contrast, *VLAs retain both strong stability and stronger adaptation to new tasks*.
> > >
> > > This result refines our claim:
> > >
> > > 1) Pretraining on diverse robot tasks improves continual learning behavior in forgetting; the multi-task training is definitely one factor in mitigating forgetting.
> > > 2) At the same time, the pretraining in VLAs retains an advantage beyond mitigating forgetting, especially in maintaining strong adaptation to new tasks (plasticity).
> > >
> > > We thank the reviewer for helping us refine the claim, and we will include the new results in the final paper revision.
> > >
> > > **4. Re. different benchmarks experiment:**
> > >
> > > We trained sequentially on RoboCasa (https://robocasa.ai/), then on LIBERO, which are very different benchmarks and vary greatly in visual and task semantics:
> > >
> > > | Training stage | Task 0 | Task 1 | Task 2 | Task 3 |
> > > |:---------------|-------:|-------:|-------:|-------:|
> > > | After RoboCasa task 0 | 0.58 | — | — | — |
> > > | After RoboCasa task 1 | 0.68 | 0.96 | — | — |
> > > | After RoboCasa task 2 | 0.38 | 0.96 | 0.44 | — |
> > > | After RoboCasa task 3 | 0.22 | 0.96 | 0.46 | 0.26 |
> > > | **After LIBERO task** | **0.22** | **1.00** | **0.64** | **0.36** |
> > >
> > > - NBT ($\downarrow$) for training on RoboCasa only: NBT = 0.044
> > >
> > > - **NBT ($\downarrow$) for training on LIBERO: NBT = −0.0046**
> > >
> > > The NBW value indicates that learning on LIBERO does not lead to forgetting of the original RoboCasa tasks.

---

### Official Review · Reviewer_8eGP · 2026-03-12

**Soundness:** 4
**Presentation:** 4
**Significance:** 3
**Originality:** 3
**Overall Recommendation:** 5
**Confidence:** 4

**Summary:**

- The paper shows that pretrained large Vision Language Action models are more resistant to forgetting than smaller policy models trained from scratch.
- They experiment with 4 models: Pi0 initialized from Paligemma (VL model) and pretrained on robotics data, Pi0 initialized from Paligemma, Pi0 trained from scratch, and BC-Transformer (small policy model)
- Pretraining is esssential in improving continual learning performance in both forward and backward transfer especially with small replay buffers
- It shows that experience replay is essential: continual learning performance with EWC and simple sequantial training both result in forgetting while experience replay is robust to forgetting.
- By swapping the VL backbone and action head, they show that forgetting occurs in the VL backbone rather than the action head.
- Most forgetting occurs when the subsequent tasks show a domain shift (diverse visual backgrounds in LIBERO-10 task), rather than when the domain are similar (LIBERO-Object) which involves simialr pick and place tasks across different objects.
- For pretrained models, the knowledge of prior tasks is still preserved in the model parameters despite degradation in performance, and can be recovered with few finetuning steps.
    - This is verified by fine-tuning the model on a prior task $k$ after it has been sequentially trained on $k$ and $k+1$. The pretrained policy reaches prior performance in a fraction of the finetuning steps, whereas policy trained from scratch requires simialar number of finetuning steps as before.

**Compliance With Llm Reviewing Policy:**

Affirmed.

**Final Justification:**

My concerns have been clarified and I recommend accepting the paper. Since my initial assessment was positive, I will maintain my score.

**Key Questions For Authors:**

Can the authors include some experiments with RL policies? Here, each sequential task is trained using RL instead of behavior cloning. Some interesting comparison would be:
- Given the same pretrained model, how does the performance of a RL policy compare to a behavior cloning policy in terms of Negative Backward Transfer?
- What's the effect of the size of the replay buffer on the performance of behavior cloning policies vs RL policies?

**Limitations:**

- While the authors do not discuss this explicitly, a limitation I see is that the experiments are performed in simulated environments. The authors recommend using experience replay, but this might not be an option in the physical world. However, this is not a huge limitation and the results in the paper are still insightful.

**Strengths And Weaknesses:**

## Strengths
- The paper provides a comprehensive analysis of the role of pretraining in continual learning for VLAs and obtains several insights:
    - It shows that pretraining is essential for knowledge retention across sequential training of tasks.
    - It shows that experience replay is essential for robust continual learning but only pretrained models can take advantage of small buffers.
- The controlled experiments are well designed and provide insightful results.
- I especially like the component swapping experiments (swapping the VL backbone and action head), that help identify the source of forgetting in the VL backbone.
- The analysis for the types of tasks that result in the most forgetting is also insightful. While these are obvious in hindsight, the paper provides a formal analysis, which is valuable.

## Weaknesses
- The paper only experiments with behavior cloning. An analysis of policies trained with Reinforcement Learning would be interesting, especially since RL has shown to obtain better performance than behavior cloning on reasoning tasks.

Overall, the paper is a well written and an insightful paper that provides a comprehensive analysis of the role of pretraining in continual learning for VLAs.

---

> ### Author Rebuttal · Authors · 2026-03-31
>
> We thank the reviewer for the positive assessment, the recognition of our controlled experimental design, and the specific appreciation of the component-swapping and task-diversity analyses. We address the questions and suggestions below.
>
> **RL policies in continual learning:** We agree that studying RL-trained policies in continual learning is an interesting direction. Our focus on behavior cloning is deliberate: current VLAs (Pi0, GR00T) are predominantly trained via imitation learning, and BC provides a clean, controlled setting to isolate the effect of pretraining on forgetting without confounding factors introduced by RL (reward shaping, exploration strategies, value function stability). We will add a discussion of this as a concrete future direction in the revision, and note the recent work on RL fine-tuning of large models [1].
>
> **Simulation vs. real-world experiments.** We acknowledge that all experiments are conducted in simulation (LIBERO). Simulation enables the controlled, large-scale comparisons across multiple models, buffer sizes, and ablations that are central to our study. Extending to real-world settings is an important next step; we are actively conducting real-robot experiments under the same sequential ER protocol. Due to time constraints, we aim to include results in the later phase of the rebuttal discussion and in the camera-ready revision.
>
> [1] Shenfeld, I., Pari, J., & Agrawal, P. (2025). RL's Razor: Why Online Reinforcement Learning Forgets Less.

---

> > ### Author Rebuttal · Reviewer_8eGP · 2026-04-01
> >
> > Thank you for your clarifications. Since my initial review was positive, I will maintain my score and recommend accepting the paper.

---

### Official Review · Reviewer_faLv · 2026-03-12

**Soundness:** 4
**Presentation:** 3
**Significance:** 3
**Originality:** 2
**Overall Recommendation:** 4
**Confidence:** 4

**Summary:**

This paper studies continual learning for pretrained VLA models and argues that such models forget much less than smaller behavior-cloning policies trained from scratch. Using LIBERO task suites, the authors show that simple experience replay yields much lower negative backward transfer for several pretrained VLAs than for BC-Transformer, and they further analyze the role of pretraining with controlled Pi0 initialization variants. The paper also presents component-swapping and fast-recovery experiments intended to support the claim that pretrained VLAs may preserve task-relevant knowledge even when task performance drops during continual fine-tuning.

**Compliance With Llm Reviewing Policy:**

Affirmed.

**Final Justification:**

During the rebuttal phase, my concerns were addressed through the authors’ clarifications.

**Key Questions For Authors:**

1. Clarification on Real-World Applicability and Sim-to-Real Gap：The authors should address whether the remarkable resistance to forgetting will hold up on physical hardware, given that real-world environments introduce unconstrained distribution shifts, sensor noise, and complex physical dynamics not present in the LIBERO benchmark. It is necessary to discuss if extremely small replay buffer sizes (e.g., 0.2%) remain sufficient for zero-forgetting in a real-world continual learning setup, or if the buffer size needs to scale with environmental complexity.
2. Elaboration on the Theoretical Mechanisms of the Phenomenon：The authors should provide more analysis explaining why large-scale pretraining fundamentally alters these learning dynamics and inherently mitigates catastrophic interference. To address this gap, adding a deeper discussion in the appendix is strongly recommended to bolster the theoretical contributions of the paper.
3. Verification of the Generalizability of Mechanistic Findings：The paper needs to address the generalizability of the component-swapping and recovery experiments, as these are currently exclusive to the Pi0 architecture. It is critical to confirm that the VL backbone remains the primary locus of forgetting across different architectures.
Overall Recommendation

**Limitations:**

The author needs to discuss the limitations of this work. The author should analyze more VLA models to ensure the generality of the conclusions.

**Strengths And Weaknesses:**

# Strengths
1. Insightful Experimental Design: By comparing three variants of the same P10 architecture (pretrained VL+Action, pretrained VL, and from-scratch), the study successfully isolates pretraining as the key causal factor for the observed resistance to forgetting.
2. Deep Dive into the Mechanisms of Forgetting: The paper moves beyond merely reporting performance metrics. The component-swapping experiment (isolating the VL backbone vs. the action head) and the recovery efficiency analysis provide deep, novel insights into where forgetting happens and how knowledge is preserved.

# Weaknesses
1. Lack of Real-World Validation: All experiments are conducted within the simulated LIBERO benchmark. A small-scale validation on a physical robot would significantly strengthen the paper's impact.
2. Limited Theoretical Explanation：While the paper excellently documents the empirical phenomenon that VLAs resist forgetting and retain knowledge, it does not delve into the underlying mechanisms. Analyzing the representational geometry or feature similarity across tasks could help explain why pretraining fundamentally alters these dynamics.
3. Limited Scope of the Ablation Studies: The paper evaluates multiple VLA architectures for the primary benchmarking, but the in-depth mechanistic analyses (pretraining isolation and component-swapping) rely entirely on the Pi0 model.

---

> ### Author Rebuttal · Authors · 2026-03-31
>
> We sincerely thank the reviewer for the thoughtful assessment and for recognizing the insightful experimental design and the depth of our mechanistic analyses. We address each concern below, with new experiments and analysis.
>
> **1. W1 & Q1: Real-World Validation and Sim-to-Real Gap**
>
> We agree that real-world validation is important and acknowledge this as a current limitation. We are actively conducting real-robot experiments under the same sequential ER protocol; due to time constraints, we aim to include results in the later phase of the rebuttal discussion and in the camera-ready revision.
>
> **2. W2 & Q2: Theoretical / Mechanistic Explanation for Why Pretraining Mitigates Forgetting**
>
> We thank the reviewer for this suggestion. We have conducted a new gradient-level analysis that provides a mechanistic perspective. We analyze the optimization dynamics using **gradient interference** [1, 2] by measuring the cosine similarity between the gradients on the current and past task data. We report the cosine similarity:
> $$\cos(g_{\text{curr}}, g_{\text{replay}}) = \frac{g_{\text{curr}} \cdot g_{\text{replay}}}{\|g_{\text{curr}}\| \|g_{\text{replay}}\|}$$ A negative cosine similarity ($\cos < 0$) indicates that updates from the new data and replay data point in opposing directions, reflecting a gradient conflict that drives continual learning interference [3].
>
> Results: Under this probe, we observe a distinct difference in optimization geometry between the two regimes. The from-scratch BC-Transformer exhibits a negative mean cosine similarity (**~-0.010**). In contrast, the pretrained VLA (Pi0) exhibits a positive mean cosine similarity (**~0.002**).
>
> Interpretation: Pretraining appears to initialize the VLA in a parameter space where joint updates on new and old-task data are inherently less antagonistic at the gradient level. Because the optimization trajectories for distinct tasks are geometrically aligned rather than in conflict, simple ER remains highly effective even with small replay budgets.
>
> References:
>
> [1] Lopez-Paz, D., & Ranzato, M. A. (2017). Gradient episodic memory for continual learning.
>
> [2] Riemer, M., et al. (2018). Learning to learn without forgetting by maximizing transfer and minimizing interference.
>
> [3] Yu, T., et al. (2020). Gradient surgery for multi-task learning.
>
> **3. W3 & Q3: Generalizability of Mechanistic Findings Beyond Pi0**
>
> We have now also conducted the recovery experiments on GR00T N1.5. The key results are qualitatively consistent with Pi0, with the finetuned version trained much faster than the task learned for the first time. The results are presented in a simplified markdown figure for readability:
>
> ```
> GR00T N1.5 Recovery Finetuning (Averaged, LIBERO-Spatial)
> SR │────────────────────────────────────────────────────────────
>  1.0│
>     │                                              ·□·····□·····□
>     │    ●●●                ·□···               ···
>     │  ●● ● ●    □·       ··     ··□···    ···□·
>  0.8│           ·  ···· ··             ·□··
>     │ ●  ●    ··       □
>     │        ·
>     │       ·
>  0.6│      □
>     │
>     │     ·
>     │    ·
>  0.4│
>     │   ·
>     │
>     │  ·
>  0.2│
>     │ ·
>     │
>     │●
>     └────────────────────────────────────────────────────────────
>      0   2k   4k   6k   8k   10k   12k   14k   16k   18k   20k
>                            Training Steps
>
>   ● ── Finetuning (recovery after learning next task)
>   □ ·· Learn First Time (original training)
>
>   GR00T recovers peak SR in ~2,400 steps (~12% of original 20,000).
> ```
>
> **4. Limitations Discussion**
>
> We will add a limitations section in the revised paper:
>
> - Sim-to-real gap: All experiments are conducted in simulation; real-world validation with sensor noise and physical dynamics are needed to confirm quantitative findings.
> - Benchmark scope: LIBERO tasks share the same embodiment and similar environments.
> - Replay buffer practicality: Experience replay assumes stored transitions from past tasks, which may face privacy or storage constraints in real-world deployments.

---

> > ### Author Rebuttal · Reviewer_faLv · 2026-04-03
> >
> > Thank you for the author's reply. Most of my questions have been answered. I look forward to your results of real-robot experiments.

---

> > > ### Author Response · Authors · 2026-04-08
> > >
> > > Thank you so much for the discussion on the real robot results!
> > >
> > > We made real robot experiments work on the mutex benchmark: https://arxiv.org/abs/2309.14320
> > >
> > > We sequentially train the pi0 model on the following three tasks using experience replay (ER), as in the previous simulation results in LIBERO. Each of these tasks comes with different visual setups in visual object layouts and task goals:
> > >
> > > - Task 0: pick up bread and put on plate
> > > - Task 1: pick up pink cup and put on plate
> > > - Task 2: pick up red cup and put into the back caddy
> > >
> > > The success rate changes after sequentially learning Task 0,1,2 are:
> > >
> > > - Task 0: 44.4% --> 47.1% --> 44.4%
> > > - Task 1: 29.4% --> 28.6%
> > > - Task 2: 33.3%
> > >
> > > SR ($\uparrow$) = $0.357$; NBT ($\downarrow$) = $−0.00183$
> > >
> > > These experiments on a physical robot, while currently on a small scale, demonstrate that continual learning behavior is also validated on real robots, with little forgetting across tasks. We will include this in our final paper revision. Thanks again for helping us include validation experiments on real hardware!

---

### Decision · Program_Chairs · 2026-04-30

**Decision:**

Accept (spotlight)

**Comment:**

This paper shows that pre-trained VLA models are better for continual learning in robotics settings when using behaviour cloning. While all the experiments in the original submission were in simulation, the authors did some real robot experiments for rebuttal. All the reviewers agree that the work is technically solid and recommend acceptance. While the idea of pre-training helping with forgetting (when using a small replay buffer) is well known in the CL community at this point, I still appreciate the demonstration of this idea in the context of robotics and VLA models.